# Hidden Markov Modeling of Reasoning Dynamics in Large Language Models

## Abstract

Reasoning in language models involves both explicit steps in the generated text and implicit structural shifts in hidden states, yet their joint dynamics remain largely underexplored. We introduce a *Explicit–Implicit Reasoning Lens (EIRL)* that jointly models these dimensions: at the explicit stage, EIRL captures transitions between reasoning roles, and at the implicit stage, it models latent depth regimes that reveal how computation is allocated across layers within each role. By linking *what* function a reasoning step serves to *where* it arises in the network, our approach provides a unified lens for both understanding reasoning dynamics and the underlying mechanisms. Once trained on reasoning trajectories, the EIRL learns probabilistic transition patterns through hidden Markov modeling that characterize how models typically move between reasoning roles and allocate computation across layers. Our analysis reveals a clear internal-to-external progression in reasoning. At the *implicit stage*, hidden states organize into distinct depth patterns that differ across reasoning categories, indicating that the model allocates its layers differently depending on the functional role of the step. These internal configurations then give rise to the *explicit stage*, where the model expresses its reasoning through semantic transitions. This progression diverges between trajectories that succeed and those that fail to reach the correct answer. Leveraging the explicit–implicit reasoning structure captured by EIRL, our framework supports both causal interventions that steer models toward targeted reasoning paths and interpretability analyses that reveal how different external intervention strategies reorganize the semantic flow of reasoning to produce their observed effects.

## 1 Introduction

Reasoning models have demonstrated strong capabilities in mathematics, scientific reasoning, and deliberative problem solving (Jaech et al., 2024; Chen et al., 2025b).Reasoning in language models manifests in two forms: explicit reasoning, which appears as semantic transitions in the generated text, and implicit reasoning, which emerges as structural shifts in hidden layer computations that reflect how the model internally allocates processing across its depth. Yet one fundamental question remain unresolved: *how can we comprehensively characterize reasoning across both its internal structural dynamics and its external semantic progression?*

Recent studies suggest that reasoning effectiveness depends not only on the correctness of individual steps but also on their structural roles, with a few "anchor" steps disproportionately shaping outcomes (Bogdan et al., 2025). Mechanistic interpretability work demonstrates that reasoning in large language models does not emerge homogeneously across all layers or components, but is carried out by specific architectural modules (Cabannes et al., 2024; Dutta et al., 2024).

Reasoning in language models runs internally-to-externally: depthwise computations shape the hidden representations that give rise to the semantic steps we observe in the generated text. To study this process systematically, we adopt a reverse-engineering perspective—starting from the external reasoning steps and tracing back to the internal structure that produces them. We introduce the *Explicit–Implicit Reasoning Lens (EIRL)*, which organizes reasoning along two stages (Figure 1). The explicit stage captures how the model moves through semantic reasoning roles in its generated text, revealing the observable flow of reasoning. The implicit stage captures how hidden states reorganize across layers within each step, exposing the structural regimes that support those semantic transi-

tions. Together, these two stages provide a unified view that links what function a step serves to where it is realized in the network, enabling a structured analysis of reasoning dynamics across both the external trajectory and the internal computational depth.

Our analysis reveals a clear internal-to-external progression in reasoning. At the *implicit stage*, hidden states organize into distinct depth patterns that differ across reasoning categories, indicating that the model allocates its layers differently depending on the functional role of the step. These internal configurations then give rise to the *explicit stage*, where the model expresses its reasoning through semantic transitions. Externally, trajectories split early into a "think-first" path that moves step by step through analysis and a "commit-early" path that jumps directly to an answer. A prominent motif is the verification loop—back-and-forth transitions between analysis and verification—which tends to delay convergence and reduce the decisiveness of the reasoning path. Taken together, these findings show that reasoning outcomes are shaped by a pipeline: internal depth dynamics set up the semantic role that appears next, and the resulting semantic transitions determine whether the trajectory advances or becomes stuck.

We validate these dynamics through both intervention and interpretability analyses. On the intervention side, EIRL provides steering vectors that shift trajectories toward successful transitions and reliably correct failing runs without increasing output length. On the interpretability side, we show that EIRL could serve as a structured tool for understanding how different intervention strategies reshape the semantic flow of reasoning—and how these changes, in turn, influence the final outcome. This provides insight into the mechanisms behind each method, rather than evaluating them solely through accuracy.

**Our contributions.** We propose EIRL, a unified framework that connects internal layer-wise computations with the external semantic stages expressed in a model's reasoning, enabling a principled analysis of reasoning dynamics. Leveraging this structure, EIRL enables both causal interventions towards targeted reasoning path and interpretability analyses of how external strategies modulate reasoning behavior.

## 2 METHODOLOGY

To systematically characterize reasoning in language models, we introduce a *Explicit–Implicit Reasoning Lens (EIRL)* that views reasoning as an internal-to-external process in which latent computation gives rise to surface text (Figure 1). Reasoning in language models runs internally-to-externally: depthwise computations shape the hidden representations that give rise to the semantic steps we observe in the generated text. To study this process systematically, we adopt a reverse-engineering perspective—starting from the external reasoning steps and tracing back to the internal structure that produces them. Within this perspective, EIRL models reasoning through two stages. At the *explicit stage*, we represent the progression of reasoning steps as transitions through semantic reasoning roles; the full category set and labeling procedure are detailed in Section 2.1. At the *implicit stage*, conditioned on each semantic role, EIRL captures the underlying layer-allocation patterns, modeled as a hidden Markov chain over depth regimes, that reveal how computation is distributed across the network for that role. Taken together, EIRL provides a principled probabilistic framework that links explicit reasoning roles with the implicit layer-depth patterns.

We segment generated tokens into reasoning steps $\{s_1, \ldots, s_T\}$ using blank-line delimiters. In parallel, we extract the hidden representations of the first token of each step, forming $H \in \mathbb{R}^{(L+1) \times T \times d}$, where $L$ is the number of transformer blocks and $d$ is the hidden dimension, with $h_{t,\ell} = H_{\ell,t,:}$ denoting the hidden state of layer $\ell$ for step $s_t$. Before modelling, these hidden vectors are standardized to zero mean and unit variance, and then projected into a $d_{pca}$-dimensional subspace using PCA. This preprocessing both stabilizes Gaussian estimation and removes redundant correlations across hidden dimensions, ensuring that the hidden markov modeling capture meaningful structural variations in depth allocation. Implementation details are in Appendix A and B.

### 2.1 EXPLICIT REASONING TRANSITIONS

At the explicit stage, we model reasoning as a sequence of semantic transitions. Prior work shows that chain-of-thought traces naturally cluster into functional phases rather than forming a uniform stream (Bogdan et al., 2025). Building on this observation, we adopt a coarse set of

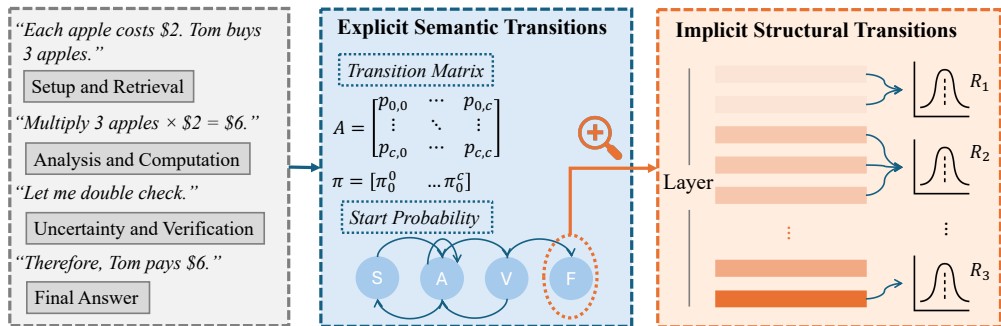

Figure 1: Overview of the *Explicit–implicit Reasoning Lens (EIRL)* workflow. At the *explicit stage*, explicit reasoning is modeled as semantic transitions across tagged reasoning steps. At the *implicit stage*, implicit reasoning is captured through latent regimes that characterize depth allocation across layers. Together, this two-stage structure links what function a step serves with where it is realized in the network, enabling systematic analysis of reasoning trajectories.

categories: `final_answer`, `setup_and_retrieval`, `analysis_and_computation`, and `uncertainty_and_verification`, plus an `unknown` fallback, that together span the full progression from problem formulation to solution emission. For each step $s_t$, we use the same model that generated the trajectory to self-classify its functional role, producing a label $c_t \in \mathcal{C}$ (details in Appendix A). The `unknown` label captures steps the model cannot clearly classify. This self-classification approach reflects the model's own assessment of each step's functional role. Given a trajectory of $T$ steps, the categorical assignments $c_{1:T}$ define a Markov chain:

$$p(c_{1:T}) = \pi_{c_1}^{0,\exp} \prod_{t=2}^{T} A_{c_{t-1},c_t}^{\exp},$$

where $\pi^{0,\exp}$ is the start distribution and $A^{\exp}$ is the transition matrix over explicit reasoning categories. The start distribution $\pi_c^{\exp}$ is estimated as the fraction of trajectories whose first labeled step is category $c$. The transition matrix $A^{\exp}$ is likewise obtained from data: $A_{i,j}^{\exp}$ is the proportion of all occurrences of category $i$ that transition to category $j$ at the next step.

This Markov representation summarizes the global flow of reasoning: high self-loop probabilities $A_{c,c}^{\exp}$ indicate stages where models tend to dwell, whereas off-diagonal entries characterize how models typically progress from one functional stage to the next.

## 2.2 IMPLICIT REASONING TRANSITIONS

At the implicit stage, we model how computation is internally allocated across layers during a reasoning step. Conditioned on the semantic category $c_t$, the PCA-projected hidden representations $\{\tilde{h}_{t,\ell}\}_{\ell=0}^{L}$ are modeled by a category-specific hidden Markov model with $K$ latent regimes. Each layer $\ell$ is assigned a discrete regime $z_{t,\ell} \in \{1, \ldots, K\}$, where consecutive layers with the same $z_{t,\ell}$ correspond to similar layer-level representation patterns. A change in $z_{t,\ell}$ marks a shift in how the network processes information across depth. For category $c$, the regime sequence follows

$$p(z_{t,0:L} \mid c_t = c) = \pi_{z_{t,0}}^{0,imp(c)} \prod_{\ell=1}^{L} A_{z_{t,\ell-1}, z_{t,\ell}}^{imp(c)},$$

where $\pi^{0,imp(c)}$ is the category-specific initial regime distribution and $A^{0,imp(c)}$ is the regime transition matrix. The observed hidden states are emitted from Gaussian distributions:

$$p(\tilde{h}_{t,\ell} \mid z_{t,\ell} = k, c_t = c) = \mathcal{N}\left(\tilde{h}_{t,\ell}; \mu_k^{(c)}, \ \mathrm{diag}(\sigma_k^{2(c)})\right),$$

where $\mu_k^{(c)}, \sigma_k^{2(c)} \in \mathbb{R}^{d_{\mathrm{pca}}}$ are regime-specific parameters.

This formulation allows the model to discover distinct layer-segmentation patterns that emerge within each stage of reasoning. Given the fixed semantic categories from Section 2.1, the parameters $\{\pi^{0,imp(c)}, A^{imp(c)}, \mu_k^{(c)}, \sigma_k^{2(c)}\}$ are learned via the Expectation–Maximization (EM) algorithm.

For each semantic category, we collect all layer-wise hidden-state sequences from steps labeled with that category. The EM procedure then alternates between (i) inferring posterior distributions over the latent regime sequence $z_{t,0:L}$ for each step using the forward–backward algorithm (E-step), and (ii) updating the regime transition probabilities and Gaussian emission parameters to maximize the likelihood of the observed PCA-projected hidden vectors $\tilde{h}_{t,\ell}$ (M-step). Through this process, layers exhibiting similar representation patterns are grouped into the same regime, whereas transitions between regimes mark points where the model shifts to a different mode of computation across depth. At inference time, the most likely regime sequence is recovered via Viterbi decoding. For example, in `final_answer` steps, the learned HMM may reveal that layers 0–10 consistently cluster into regime $z = 1$ (characterized by one Gaussian distribution), layers 11–25 transition to regime $z = 2$ with a distinct distribution, and layers 26–32 shift to regime $z = 3$. These segmentation patterns—i.e., which layers are assigned to which regime—emerge directly from fitting the HMM to the observed hidden-state trajectories. They provide a data-driven picture of how computation is organized across network depth during different stages of reasoning.

The EIRL framework provides a structured lens on reasoning dynamics: semantic transitions capture *what* stages of reasoning unfold and how they progress, while structural regimes capture *how* computation is internally organized within each stage.

## 3 EXPERIMENTAL SETUP

**Models.** We evaluate four open-source reasoning models spanning different parameter scales (1.7B–7B) and architectures: Bespoke-Stratos-7B (Labs, 2025), OpenThinker-7B (Guha et al., 2025), Qwen3-1.7B (Team, 2025), and Llama-3.1-Nemotron-Nano-4B-v1.1 (Bercovich et al., 2025). Bespoke-Stratos and OpenThinker are both distilled from DeepSeek-R1 on different datasets using Qwen2.5-7B-Instruct as the base, enabling comparison of distillation effects. Qwen3 shares the Qwen architecture but uses a different training recipe, while Nemotron represents the LLaMA lineage. This selection disentangles effects of model size, architecture, and training methodology.

**Datasets.** Our experiments span four benchmarks varying in difficulty and domain: MATH-500 (Lightman et al., 2023) (mathematics), GPQA-Diamond (Rein et al., 2024) (graduate-level science), WebInstruct-Verified (Ma et al., 2025) (diverse real-world problems), and AIME-2024 (Hugging-FaceH4, 2024) (competition mathematics). This spectrum, from standard problem-solving to expert-level challenges, makes our findings generalize across difficulty levels and domains.

## 4 EXPLICIT STAGE REASONING DYNAMICS

To understand how language models reason, we focus on how their trajectories unfold, how paths leading to correct predictions differ from those not, and how these patterns vary across models. All subsequent analyses focus on the four core reasoning categories, excluding *unknown*, which captures ambiguous steps where the model's functional intent is unclear.

*Q1. How do reasoning trajectories flow?*

Reasoning does not proceed randomly: it follows clear, repeatable patterns. Table 1 presents the transition probabilities averaged across all models and datasets. For clarity in the following analysis, we use abbreviated labels for the four reasoning categories: Setup (setup_and_retrieval), Analysis (analysis_and_computation), Verify (uncertainty_and_verification), and Final (final_answer). Transition patterns reveal three motifs: stable stages with self-loops, branching points where reasoning diverges, and verification loops that revisit analysis.

**i) Stable stages** (blue in Table 1). The diagonal entries of the transition matrix represent self-loop probabilities: the likelihood that a reasoning step in category $i$ is followed by another step in the same category. *Analysis_and_computation* and *final_answer* exhibit the highest self-loop probabilities (0.505 and 0.497 respectively), indicating that once the model enters these stages, it typically remains there for multiple consecutive steps. In contrast, *setup_and_retrieval* (0.323) and *uncertainty_and_verification* (0.267) show lower self-transition rates, meaning the model quickly moves on from these stages.

**ii) Branching point** (yellow in Table 1). The *setup_and_retrieval* row shows balanced outgoing transitions to multiple destinations, making it a major decision point. From setup, the model may transition to *analysis_and_computation* ($A^{\text{exp}}_{\text{setup,analysis}} = 0.355$), jump directly to *final_answer* ($A^{\text{exp}}_{\text{setup,final}} = 0.245$), or remain in setup ($A^{\text{exp}}_{\text{setup,setup}} = 0.323$) to gather more information. These three pathways correspond to distinct reasoning styles: deliberate reasoning ("think-first"), shortcut answering ("commit-early"), and repeated information gathering ("re-setup").

Table 1: Average transition probabilities across all models and datasets. Rows represent source categories, columns represent destination categories. The matrix reveals three key patterns: strong self-loops indicating stable stages (analysis, final answer), branching points where reasoning diverges (setup), and verification loops.

|  | Final | Setup | Analysis | Verify |
|---|---|---|---|---|
| Final | 0.497 | 0.111 | 0.272 | 0.117 |
| Setup | 0.245 | 0.323 | 0.355 | 0.076 |
| Analysis | 0.245 | 0.132 | 0.505 | 0.116 |
| Verify | 0.222 | 0.168 | 0.362 | 0.267 |

**iii) Verification loop** (green in Table 1). The *uncertainty_and_verification* stage functions primarily as a temporary check. The off-diagonal entries show that models frequently transition from *analysis_and_computation* to *verification* (0.116) and back (0.362), forming a characteristic loop pattern. Additionally, models occasionally return to verification after producing an answer (*final_answer* → *verification*, 0.117), suggesting a tendency to double-check conclusions before or after committing to a final response.

### Q2. What distinguishes trajectories that lead to correct predictions (successful) from those that fail to reach correct predictions (unsuccessful)?

Table 2 compares the transition patterns of trajectories that ultimately reach correct predictions versus those that do not, averaged across all models and datasets. Both groups begin predominantly in *setup_and_retrieval* (start probabilities: 0.458 vs. 0.490). Successful trajectories then progress efficiently through analysis ($A^{\text{exp}}_{\text{setup,analysis}} = 0.387$) and then transitions decisively to final ($A^{\text{exp}}_{\text{analysis,final}} = 0.276$). The full two-step probability is therefore: $P_{\text{succ}}(\text{setup} \rightarrow \text{analysis} \rightarrow \text{final}) = 0.387 \times 0.276 = 0.1067$. Once in *final*, successful trajectories stabilize strongly ($A^{\text{exp}}_{\text{final,final}} = 0.515$). Unsuccessful trajectories chart a different progression. They linger in setup ($A^{\text{exp}}_{\text{setup,setup}} = 0.328$) and enter analysis with a lower probability (0.346). Even when following the analogous two-step path, $P_{\text{unsucc}}(\text{setup} \rightarrow \text{analysis} \rightarrow \text{final}) = 0.346 \times 0.231 = 0.0798$, a 33.7% relative drop from successful trajectories.

Table 2: Comparison of start distributions and transition probabilities between trajectories that reach correct predictions and those that do not. Successful trajectories show clear forward transitions (setup → analysis → final) with strong stabilization. Unsuccessful trajectories dwell in setup or analysis, cycle with verification, and exhibit weaker convergence. Colored cells highlight pathways.

|  | Successful | | | | | Unsuccessful | | | | |
|---|---|---|---|---|---|---|---|---|---|---|
|  | Start Prob | Transition Matrix | | | | Start Prob | Transition Matrix | | | |
|  |  | Final | Setup | Analysis | Verify |  | Final | Setup | Analysis | Verify |
| Final | 0.087 | 0.515 | 0.110 | 0.252 | 0.123 | 0.082 | 0.471 | 0.112 | 0.298 | 0.118 |
| Setup | 0.458 | 0.243 | 0.300 | 0.387 | 0.068 | 0.490 | 0.245 | 0.328 | 0.346 | 0.080 |
| Analysis | 0.321 | 0.276 | 0.128 | 0.493 | 0.101 | 0.268 | 0.231 | 0.133 | 0.512 | 0.121 |
| Verify | 0.015 | 0.246 | 0.158 | 0.359 | 0.236 | 0.016 | 0.181 | 0.171 | 0.367 | 0.279 |

Once in *analysis*, the two groups diverge sharply. Successful trajectories typically pass through analysis only briefly and exit toward the final answer ($A^{\text{exp}}_{\text{ana,final}} = 0.276$). In contrast, unsuccessful trajectories show two forms of stagnation. First, they have a stronger analysis self-loop (0.512 vs. 0.493). Even though this step is small in magnitude, the per-step discrepancy compounds quickly: for example, over ten analysis steps, $0.493^{10} \approx 0.0067$ vs. $0.512^{10} \approx 0.0127$, making unsuccessful trajectories nearly twice as likely to remain stuck in analysis once they enter it. Second, unsuccessful trajectories are substantially more likely to fall into a full *analysis* → *verification* → *analysis* (AVA) oscillation. The probability of one AVA cycle is: $P_{\text{succ}} = 0.101 \times 0.359 = 0.0362$, $P_{\text{unsucc}} = 0.121 \times 0.367 = 0.0444$, meaning unsuccessful trajectories enter this oscil-

latory loop 22.7% more often. This AVA loop reinforces hesitation: the model repeatedly reevaluates partial reasoning instead of advancing toward a conclusion. Together, the amplified analysis self-loop and the higher-probability AVA oscillation explain why unsuccessful trajectories linger in mid-level reasoning, while successful trajectories move decisively through setup → analysis → final with stronger convergence. Statistical significant analysis see Appendix G.

***Q3: Why do models diverge in their reasoning paths?***

Divergence arises from differences in architecture and training lineage (Table 3).

**Architectural tendencies.** Nemotron (LLaMA-based) displays a strongly *direct-closure* pattern. It transitions from setup directly to final with high probability ($A^{\text{exp}}_{\text{setup,final}} = 0.652$), and once in `final_answer`, it remains there with strong stability ($A^{\text{exp}}_{\text{final,final}} = 0.638$). Qwen3 exhibits the opposite tendency: a *deliberative, verification-centric* pattern. It moves from analysis into verification at a higher rate ($A^{\text{exp}}_{\text{analysis,verify}} = 0.126$) and is strongly pulled back from verification into analysis ($A^{\text{exp}}_{\text{verify,analysis}} = 0.405$). Qwen3 also revisits setup more often than other models ($A^{\text{exp}}_{\text{verify,setup}} = 0.262$), forming extended recursion cycles. The two-step verification loop has probability $P_{\text{Qwen3}}(\text{analysis} \to \text{verify} \to \text{analysis}) = 0.126 \times 0.405 = 0.051$. In contrast, Nemotron's corresponding two-step loop is far weaker, $P_{\text{Nemo}}(\text{analysis} \to \text{verify} \to \text{analysis}) = 0.039 \times 0.309 = 0.012$, a more than four-fold reduction compared to Qwen3.

**Training lineage.** Stratos and OpenThinker—both distilled from DeepSeek-R1 with the same Qwen2.5-7B-Instruct base—show closely aligned transition patterns. Compared to Qwen3, Stratos and OpenThinker differ mainly in Qwen3's stronger verification recursion—particularly its higher verify→setup transition (0.262 vs. 0.165/0.148). Yet despite this extra pull-back, their AVA loops remain similar, and overall the three models still fall within the same Qwen-style family of deliberative reasoning. In contrast, both Stratos and OpenThinker differ sharply from Nemotron.

Table 3: Key transition statistics across models. Nemotron exhibits decisive flow with strong forward transitions; Qwen3 shows recursive patterns with intensified verification loops.

| Model | Setup→Analysis | Setup→Final | Analysis→Verify | Verify→Analysis | Verify→Final | Verify→Setup | Analysis Loop | Final Loop |
|---|---|---|---|---|---|---|---|---|
| Nemotron | 0.189 | **0.652** | 0.039 | 0.309 | **0.511** | 0.095 | 0.505 | **0.638** |
| Qwen3 | 0.412 | 0.098 | **0.126** | **0.405** | 0.068 | **0.262** | 0.142 | 0.367 |
| Stratos | 0.400 | 0.130 | 0.154 | 0.349 | 0.110 | 0.165 | 0.170 | 0.454 |
| Openthinker | 0.419 | 0.100 | 0.147 | 0.384 | 0.119 | 0.148 | 0.164 | 0.530 |

# 5 IMPLICIT STAGE REASONING DYNAMICS

So far, we have examined *semantic transitions*. We now turn to the *internal structure*: how models allocate computation across layer depth. For each reasoning step, Viterbi decoding assigns each layer to one of $K = 7$ regimes. Transitions between regimes mark regime boundaries: points where the hidden-state trajectory shifts into a region better explained by a different emission distribution. We identify stable boundaries by first grouping consecutive layers that share the same regime (assigned by the mode computed across all samples and steps), and then voting across datasets and seeds to retain only those boundaries that appear in $\geq 25\%$ of runs. We normalize boundaries to same scale to compare cross-model similarity, details in Appendix E. Figure 2 reveals that models cluster by training lineage and architectural families: Stratos and OpenThinker exhibit near-perfect alignment (similarity $\approx 0.8$–$1.0$), reflecting their shared distillation from R1, while Nemotron diverges substantially (similarity $\approx 0.2$–$0.3$). Therefore, we analyze each family separately: Qwen-family (Stratos, OpenThinker, Qwen) and LLaMA-family (Nemotron).

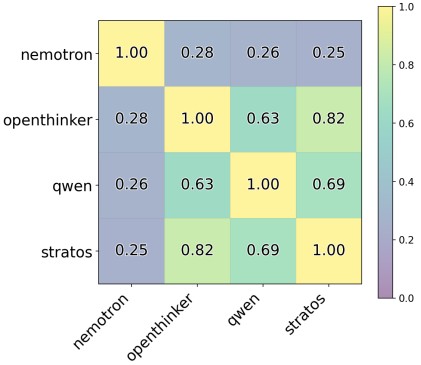

Figure 2: Cross-model similarity in boundary placement.

Table 4: Family-level boundaries where the *uncertainty_and_verification* step shows different pattern, with Jaccard score(mean ± std across models) measures boundary position overlap between successful and unsuccessful trajectories.

| Family | Category | Segments | Jaccard |
|---|---|---|---|
| Qwen | setup | 11 segments: 0–10, 11–18, 19, 20, 21, 22, 23, 24, 25, 26–27, 28 | 0.71 (0.64–0.83) |
| | analysis | 10 segments: 0–7, 8–18, 19, 20–21, 22, 23, 24, 25, 26–27, 28 | 0.66 (0.50–0.80) |
| | verify | 11 segments: 0–17, 18, 19, 20, 21, 22, 23, 24, 25, 26–27, 28 | 0.56 (0.46–0.64) |
| | final | 9 segments: 0–7, 8–17, 18–19, 20, 21, 22–24, 25, 26–27, 28 | 0.56 (0.46–0.73) |
| LLaMA | setup | 11 segments: 0–9, 10, 11–12, 13–16, 17–18, 19, 20–22, 23–26, 27, 28–31, 32 | 0.56 |
| | analysis | 9 segments: 0–10, 11, 12–17, 18, 19–25, 26, 27–28, 29–31, 32 | 0.56 |
| | verify | 7 segments: 0–12, 13–16, 17, 18–20, 21–23, 24–31, 32 | 0.38 |
| | final | 11 segments: 0–9, 10, 11, 12–16, 17, 18–22, 23–25, 26, 27, 28–31, 32 | 0.17 |

### *Q1. How do reasoning categories differ in depth allocation?*

Table 4 presents the family-level boundaries aggregated across all trajectories. Different reasoning categories exhibit distinct depth allocation patterns. *Uncertainty_and_verification* displays a *delayed first boundary* pattern in Qwen-family: its first segment spans layers 0–17 before the next regime transition occurs. This suggests that during verification steps, the model's hidden representations remain within a single Gaussian cluster across early-to-mid layers, with differentiation occurring only in upper layers. LLaMA-family shows similar but less pronounced delay.

### *Q2. Do trajectories leading to correct predictions (successful) use depth similarly to those that fail (unsuccessful)?*

We quantify structural divergence between successful and unsuccessful trajectories using Jaccard similarity: $J(c) = |\mathcal{B}^{(c)}_{\text{succ}} \cap \mathcal{B}^{(c)}_{\text{unsucc}}| / |\mathcal{B}^{(c)}_{\text{succ}} \cup \mathcal{B}^{(c)}_{\text{unsucc}}|$, where $\mathcal{B}^{(c)}_{\text{succ}}$ and $\mathcal{B}^{(c)}_{\text{unsucc}}$ denote consensus boundary positions (excluding fixed endpoints) for category $c$. This metric equals 1 when successful and unsuccessful trajectories place boundaries at identical depths and 0 when they share none. Table 4 reveals that *setup_and_retrieval* shows the high boundary overlap, indicating that early reasoning structure is relatively preserved even in failing trajectories. In contrast, *uncertainty_and_verification* exhibits the different behavior, suggesting that successful and unsuccessful trajectories organize verification steps most differently.

## 6 Intervening on Semantic Boundaries

Our analysis have shown that reasoning follows structured semantic stages. We now ask whether modifying a model's internal state according to the EIRL-inferred reasoning transition pattern can redirect its reasoning path and thereby change its reasoning behavior and output. To test this, we derive *steering vectors* from the EIRL transition matrix and inject them into the model's hidden states at selected transition layers. If such interventions alter or improve reasoning outcomes, it would suggest that these transitions act as functional control points within the model's computation.

**Edge-conditioned Displacement Vectors.** We construct steering vectors that target specific semantic transitions(edges) between reasoning categories, where an edge $(a \rightarrow b)$ represents a transition from category $a$ to category $b$. Let $\Phi : \mathbb{R}^d \rightarrow \mathbb{R}^k$ denote the preprocessing from §2. For each step $s_t$, we map the $\ell^{th}$ layer hidden state $h_{t,\ell} \in \mathbb{R}^d$ to $\tilde{h}_{t,\ell} = \Phi(h_{t,\ell}) \in \mathbb{R}^k$, and for each transition $(c_t, c_{t+1})$, we compute the displacement $d_t = \tilde{h}_{t+1,\ell} - \tilde{h}_{t,\ell}$. Separating displacements by trajectory outcome, we average them per edge: for successful trajectories, $\mu^{\text{s}}_{a \rightarrow b}$ averages all $d_t$ along $(a \rightarrow b)$; for unsuccessful ones, $\mu^{\text{us}}_{a \rightarrow b}$ averages displacements from source $a$ to all destinations except $b$:

$$\mu^{\text{s}}_{a \rightarrow b} = \frac{1}{N^{\text{s}}_{a \rightarrow b}} \sum_{(t:\, y=\text{s},\, c_t=a,\, c_{t+1}=b)} d_t, \quad \mu^{\text{us}}_{a \rightarrow b} = \frac{1}{N^{\text{us}}_a - N^{\text{us}}_{a \rightarrow b}} \sum_{t:\, y=\text{us},\, c_t=a,\, c_{t+1}\neq b} d_t.$$

The $\Delta_{a \to b}$ is computed as $\mu_{a \to b}^{\mathsf{s}} - \mu_{a \to b}^{\mathsf{us}} \in \mathbb{R}^k$, and then normalized. This allows us to capture how the model's internal trajectory diverges when reasoning fails. To perform steering in the model's hidden space, we inverse preprocess $\Phi$ and map $v_{a \to b} = \Phi^{-1}(\Delta_{a \to b}) \in \mathbb{R}^d$. See Algorithm 1.

**Edge selection and steering methods.** For explicit transition $(a \to b)$, we compute its divergence $\Delta_{a,b} = A_{a,b}^{\exp}(\text{succ}) - A_{a,b}^{\exp}(\text{unsucc})$, and rank edges by $\Delta_{i,j}$. We evaluate two strategies. *Hard steering* applies the vector derived from the single edge with the largest $\Delta_{i,j}$. *Soft steering* samples among the top-3 edges using softmax-normalized weights $p_{i,j} \propto \exp(\Delta_{i,j})$. We compare these against an *edge-agnostic* baseline that uses one global vector derived from the overall correct–incorrect contrast averaged all steps.

At generation time, we apply steering at step boundaries (detected by blank-line delimiters): $h_{t,\ell} \leftarrow h_{t,\ell} + \alpha\, v_{a \to b}$, where $\ell$ is the intervention layer and $\alpha$ controls strength. We test two sites: (a) the *final layer* ($\ell = L$), where §5 shows models consistently place regime boundaries, and (b) the *transition layer*: the layer where the EIRL detects a regime change. For each semantic category, we identify the consensus regime at each layer by taking the mode across all samples. The consensus layer set consists of the median transition layer across all categories, where transition layers mark distribution changes in the implicit stage of EIRL modeling.

Table 5: Correction rate ($\uparrow$; fraction of originally incorrect predictions corrected after steering) and token count ($\downarrow$; token counts of originally incorrect predictions), averaged over 3 independent runs. Steer at the last layer.

| Model | Dataset | Steering Correction Rate $\uparrow$ | | | Tokens $\downarrow$ | | |
|---|---|---|---|---|---|---|---|
| | | Edge-Agnostic | Soft | Hard | Baseline | Soft | Hard |
| Bespoke-Stratos-7B | MATH-500 | 0.2515 | 0.2697 | 0.2212 | 1970 | 1849 | 1845 |
| | WebInstruct-Verified | 0.1632 | 0.1875 | 0.1597 | 2938 | 2944 | 2953 |
| | GPQA-Diamond | 0.1500 | 0.1682 | 0.1773 | 3927 | 3781 | 3848 |
| OpenThinker-7B | MATH-500 | 0.2126 | 0.2216 | 0.2126 | 1995 | 1949 | 1960 |
| | WebInstruct-Verified | 0.1448 | 0.1448 | 0.1345 | 3905 | 3804 | 3765 |
| | GPQA-Diamond | 0.1181 | 0.1266 | 0.1561 | 4700 | 4575 | 4483 |
| Qwen3-1.7B | MATH-500 | 0.1506 | 0.1446 | 0.1627 | 1997 | 1989 | 1982 |
| | WebInstruct-Verified | 0.1241 | 0.1448 | 0.1207 | 3793 | 3703 | 3787 |
| | GPQA-Diamond | 0.0317 | 0.0357 | 0.0516 | 4731 | 4649 | 4604 |
| Llama-3.1-Nemotron-Nano-4B-v1.1 | MATH-500 | 0.1562 | 0.1622 | 0.1742 | 1980 | 1959 | 1954 |
| | WebInstruct-Verified | 0.0951 | 0.1344 | 0.1279 | 4110 | 4061 | 4099 |
| | GPQA-Diamond | 0.0565 | 0.0605 | 0.0524 | 4909 | 4793 | 4786 |
| | **Average** | 0.1379 | 0.1500 | 0.1459 | 3413 | 3338 | 3339 |

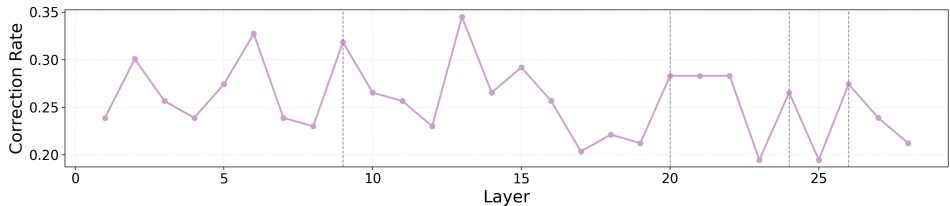

Figure 3: Layer-wise hard steering performance on MATH-500 using Bespoke-Stratos-7B (seed=1). Dashed vertical lines indicate consensus transition layers (9, 20, 24, 26). Steering at these transition layers consistently shows a local optima in performance.

**Targeted interventions improve accuracy without added tokens.** Table 5 summarizes correction rates and token counts for both edge-agnostic and edge-conditioned interventions. Averaging across models and datasets, edge-conditioned steering consistently improves correction rates without increasing output length. A qualitative sample is provided in Appendix H to illustratively show the steering effect. Interestingly, Figure 3 reveals an alignment between EIRL-identified transition layers and steering performance. The dashed vertical lines mark consensus transition layers—layers

where the hidden layer regime changes identified by implicit-stage EIRL. Performance consistently reaches a local optima at these layers, demonstrating that EIRL successfully identifies critical transition points in the model's internal reasoning process. While other layers occasionally show competitive performance (possibly due to suboptimal regime granularity in our $K$=7 setting), the consistent alignment between EIRL transitions and performance ridges validates that these layers capture genuine shifts in information processing. Together, this suggests that the EIRL provides insights for both internal depth organization and external semantic transitions act as functional control points: targeted interventions at transition layer using reasoning-type transitions identified by EIRL can redirect failing trajectories toward correct predictions, suggesting that intervening at these layers can guide reasoning to targeted trajectory and thereby change the final outcome.

***Uncertainty_and_verification* to *final_answer* transitions are central to success.** To identify which transitions matter most, we analyze the most frequent top-1 edges across models (Table 6). Globally, the dominant transition is *uncertainty_and_verification* → *final_answer*, shared by Nemotron, Stratos, and OpenThinker. Qwen instead relies on *setup_and_retrieval* → *analysis_and_computation* and *analysis_and_computation* → *final_answer*. This concentration reveals that models succeed through decisive transitions, typically resolving uncertainty or concluding analysis.

Table 6: Most common top-1 semantic transitions across models and datasets.

| Scope | Top-1 Edge |
|---|---|
| Global | uncertainty_and_verification → final_answer |
| Nemotron | uncertainty_and_verification → final_answer |
| OpenThinker | uncertainty_and_verification → final_answer |
| | setup_and_retrieval → analysis_and_computation |
| | and analysis_and_computation → final_answer |
| Qwen | |
| Stratos | uncertainty_and_verification → final_answer |

## 7 USING EIRL AS AN INTERPRETABILITY TOOL FOR REASONING INTERVENTION STRATEGIES

A wide range of intervention strategies aim to influence reasoning outcomes, typically seeking higher accuracy or shorter outputs. However, these strategies are often assessed only through aggregate metrics, offering little insight into how they alter the semantic flow of reasoning and why those alterations lead to different outcomes. EIRL provides an interpretability and diagnostic framework for this setting: rather than relying solely on end metrics, it reveals how each strategy reshapes the reshape the semantic flow of reasoning and explains how these changes, in turn, influence the final outcome. Concretely, we compare three prompting strategies: *P1* is our baseline prompt (full text in Appendix A); *P2* augments P1 with the instruction "Be concise"; *P3* further adds a direction-guided sentence that explicitly instructs the model to reason in a designed forward manner: *"Reason in a clear forward direction, moving from the setup stage to the analysis stage and finally to the answer stage, without looping back or re-checking previous steps. Avoid unnecessary verification and commit to a single consistent line of reasoning."* Results are in Table 7.

Table 7: Start probabilities and explicit stage transition matrices for the three prompting strategies on Bespoke-Stratos-7B (MATH-500). Together with the strategy's accuracy and average token count.

| | P1: Baseline CoT | | | | P2: + Concise | | | | P3: Direction-guided | | | |
|---|---|---|---|---|---|---|---|---|---|---|---|---|
| | Final | Setup | Analysis | Verify | Final | Setup | Analysis | Verify | Final | Setup | Analysis | Verify |
| Start | 0.000 | 0.708 | 0.280 | 0.012 | 0.000 | 0.656 | 0.328 | 0.016 | 0.000 | 0.620 | 0.348 | 0.032 |
| Final | 0.493 | 0.071 | 0.274 | 0.162 | 0.503 | 0.085 | 0.267 | 0.145 | 0.510 | 0.061 | 0.267 | 0.162 |
| Setup | 0.110 | 0.342 | 0.487 | 0.061 | 0.106 | 0.298 | 0.519 | 0.077 | 0.089 | 0.312 | 0.535 | 0.064 |
| Analysis | 0.177 | 0.159 | 0.567 | 0.096 | 0.157 | 0.139 | 0.608 | 0.095 | 0.157 | 0.135 | 0.591 | 0.116 |
| Verify | 0.141 | 0.220 | 0.388 | 0.250 | 0.106 | 0.218 | 0.413 | 0.263 | 0.123 | 0.178 | 0.438 | 0.261 |
| Acc / Tok | 0.560 / 1505 | | | | 0.588 / 1367 | | | | 0.576 / 1492 | | | |

we use these EIRL transitions in Table 7 to diagnose how each strategy restructures the progression of reasoning and how such restructuring explains their differing accuracies and output lengths. All settings begin predominantly in setup_and_retrieval, but both P2 and P3 noticeably reduce this starting probability ($0.708 \rightarrow 0.656/0.620$). This confirms the intuitive function of these prompts: both push the model to start working sooner instead of restating the question.

*P1 → P2: Concise prompting reduces loops and stabilizes reasoning.* Compared to P1, P2 increases the probability of moving directly into analysis (Setup→Analysis: 0.487 → 0.519) and strengthens the analysis self-loop (0.567 → 0.608), while slightly lowering transitions into verification. This produces a more decisive progression with fewer back-and-forth cycles, explaining the accuracy improvement and the reduction in token count. In short, P2 compresses the trajectory by reducing redundancy and converging more reliably to the answer.

*P2 → P3: Direction guidance introduces structured yet only partially reliable control.* Relative to P1, P3 strengthens forward movement (Setup→Analysis: 0.487 → 0.535) and increases Verify→Analysis (0.388 → 0.438), showing that once verification occurs, the model now returns to analysis more quickly. At the same time, Analysis→Verify becomes more frequent (0.096 → 0.116), indicating that verification is still triggered often. Thus, prompt-level guidance can steer the reasoning flow toward the intended forward structure, but the effect remains partial.

This highlights the role of EIRL as an interpretability diagnostic tool: it reveals not just whether an intervention changes performance, but how it reshapes the underlying semantic transitions.

## 8 RELATED WORK

Recent studies highlight that reasoning models can exhibit unintended or deceptive behaviors, underscoring the need for a deeper mechanistic understanding (Baker et al., 2025). Several works examine which aspects of CoT steps matter. Madaan & Yazdanbakhsh (2022) disentangle the roles of textual content and structural patterns, Wang et al. (2022) show that performance gains often persist even with flawed step content, and Bogdan et al. (2025) find that a small set of "anchor" steps disproportionately shape final outcomes.

Mechanistic analyses probe the internal processes behind reasoning. Cabannes et al. (2024) and Dutta et al. (2024) show how architectural components enable stepwise reasoning. Latent multi-hop phenomena are revealed by Yang et al. (2024b) and Shalev et al. (2024), while Venhoff et al. (2025) demonstrate that steering vectors can capture functional directions in hidden space. Information-theoretic perspectives provide a complementary lens: Ton et al. (2024) analyze CoT dynamics through entropy and mutual information, while Punjwani & Heck (2025) explore how neural network weights encode and constrain reasoning capacity. Beyond analysis, several works pursue interventions. Chen et al. (2025a) introduce training-free latent steering to suppress over-reflection, while Venhoff et al. (2025) show that representation-level modifications can modulate reasoning style. Wang et al. (2025b) propose efficient post-training refinement of latent reasoning, enabling reasoning improvements without full retraining. Reasoning efficiency survey (Sui et al., 2025) catalog methods to mitigate "overthinking," and token-level analyses (Wang et al., 2025a) identify sparse high-entropy tokens as critical intervention points. Two concurrent works explicitly model state-aware dynamics of reasoning. Wu et al. (2025) frame CoT as a latent-state MDP, training a transition policy with RL to improve reasoning exploration. Yu et al. (2025) cluster final-layer embeddings of steps and construct a Markov chain to visualize reasoning motifs. Our work differs by introducing a *two-stage HMM* that integrates explicit semantic roles with hidden layer regimes. This design ties *what* function a step serves to *where* it arises in the network, and further enables *EIRL-informed* intervention and diagnostics.

## 9 CONCLUSION

We introduced *Explicit–implicit Reasoning Lens (EIRL)* that integrates semantic reasoning roles with latent depth regimes, linking *what* a step does to *where* it arises in the network. This two-stage perspective reveals a clear internal-to-external progression in reasoning. At the implicit stage, hidden states organize into distinct depth patterns that differ across reasoning categories, indicating that the model allocates its layers differently depending on the functional role of the step. These internal configurations then give rise to the explicit stage, where the model expresses its reasoning through semantic transitions. Beyond providing an explanatory lens, EIRL supports both causal control—via transition-conditioned steering that reliably rescues failing runs without lengthening outputs—and diagnostic analysis, revealing how interventions reorganize reasoning flow and produce their performance effects. Taken together, these results position EIRL as both a framework for understanding and reshaping reasoning.

**Ethics Statement.** This work adheres to the Code of Ethics. Our experiments use only open-source models and publicly available datasets under their respective open licenses, with no involvement of human subjects or sensitive data. We identify no foreseeable ethical risks.

**Reproducibility Statement.** We ensure reproducibility by providing experimental and implementation details in Section 3 and Appendices A–E. Full results with statistical significance are in Appendices C–E, and anonymous source code is included as supplementary material.

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

# A  DATA PREPROCESSING AND EXPERIMENTAL SETUP

## A.1  MODELS

We evaluate five open-source reasoning models, each paired with a classification model where applicable. All models are run with bfloat16 precision unless otherwise specified. Generation models use standard reasoning hyperparameters, while classification models are configured with do_sample = False. Licenses are listed for reproducibility.

*Qwen3-1.7B* (Team, 2025) was used for both generation and classification, with "thinking mode" enabled for generation and disabled for classification. The parameters were: temperature = 0.6, top_p = 1.0, top_k = 20, and min_p = 0; classification was run deterministically with do_sample = False. License: Apache 2.0.

*Bespoke-Stratos-7B* (Labs, 2025) was paired with *Qwen2.5-7B-Instruct* (Yang et al., 2024a) for classification. The configuration used temperature = 0.6, top_p = 0.95, and deterministic classification (do_sample = False). License: Apache 2.0 (both models).

*OpenThinker-7B* (Guha et al., 2025) was paired with *Qwen2.5-7B-Instruct* (Yang et al., 2024a), using the same settings (temperature = 0.6, top_p = 0.95, and do_sample = False for classification). License: Apache 2.0 (both models).

*Llama-3.1-Nemotron-Nano-4B-v1.1* (Bercovich et al., 2025) was used for both generation and classification. "Thinking mode" was enabled for generation and disabled for classification, with parameters temperature = 0.6, top_p = 0.95, and deterministic classification (do_sample = False). License: NVIDIA Open Model License.

Besides above models, LLaMA-3.1-8B-Instruct (Meta, 2024) are used only for WebInstruct-Verified, with Meta LLaMA 3.1 License.

## A.2  DATASETS

Our experiments span 4 benchmarks. Splits, sizes, maximum token cutoffs and licenses are shown in Table 8. AIME-2024 is excluded from steering analysis due to its small size and difficulty.

Table 8: Datasets used in experiments.

| Dataset | Split | Train Size | Test Size | Max Tokens | License |
|---|---|---|---|---|---|
| MATH-500 (Lightman et al., 2023) | test | 250 | 250 | 2000 | MIT |
| GPQA-Diamond (Rein et al., 2024) | test | 100 | 98 | 5000 | Apache 2.0 |
| WebInstruct-Verified (Ma et al., 2025) | test | 144 | 144 | 5000 | Apache 2.0 |
| AIME-2024 (HuggingFaceH4, 2024) | train | 15 | – | 32768 | Apache 2.0 |

For WebInstruct-Verified, we additionally filter by: answer_types as Float, Multiple Choice, Integer, Percentage and difficulties as Primary, Junior High, Senior High.

## A.3  PROMPTING

**Generation.**  - **Multiple-choice tasks (GPQA)** use:

```
You are answering a multiple-choice question.
Options are labeled A, B, C, and D.
Think step-by-step and show your reasoning.
At the very end, output ONE line exactly in this format:
Final Answer: \boxed{A}
```

- **Open-ended tasks (MATH-500, AIME-2024, WebInstruct)** use:

```
Answer the following question step-by-step.
At the very end, output exactly one line formatted as:
Final Answer: \boxed{...}
```

**Classification.** For step-level semantic tagging, all classification models use:

```
You are an expert in reasoning analysis.
Classify the function of each sentence into one of the following tags:
1. final_answer
2. setup_and_retrieval
3. analysis_and_computation
4. uncertainty_and_verification
```

If the classification model cannot assign a tag to the step, it is labeled as unknown.

## A.4 PREPROCESSING PIPELINE

We segment each generated solution into reasoning steps using blank-line delimiters, and for every step we store the hidden representation of its first token across all layers as the step-level hidden state. Per-step reasoning sequences are constructed and anchored to semantic labels. Across all runs, reasoning steps are distributed as: analysis_and_computation (40.2%), final_answer (28.0%), setup_and_retrieval (17.5%), uncertainty_and_verification (13.9%), and unknown (0.3%).

## A.5 COMPUTING

All experiments were conducted with NVIDIA H100 and H200 GPUs.

# B EIRL CONFIGURATIONS

## B.1 TRAINING

**Data Loading and Filtering** Classification labels and step-level hidden states (extracted during Appendix A) are loaded. Depending on the configuration, training uses either all samples, only correct samples, or only incorrect samples.

**Feature Preprocessing** Step embeddings are standardized with a StandardScaler, then reduced via PCA to a target dimension of 64 ($d_{pca} = 64$). PCA is run with the solver option svd_solver=full.

**EIRL Fitting** The EIRL is trained with 5 fixed explicit-level categories ($C = 5$) aligned to semantic anchors, and seven Implicit stage regimes ($K = 7$) per category. Training runs for 10 EM iterations (n_iter = 10).

**Anchored Explicit-stage Categories** Explicit-stage categories are fixed by semantic labels. Each reasoning step is assigned one of 5 canonical categories: final_answer, setup_and_retrieval, analysis_and_computation, uncertainty_and_verification, unknown. These labels are required for every sequence and are coerced into integer IDs (0–4). The explicit stage start and transition probabilities are estimated directly from observed label sequences.

**Implicit Stage HMMs** For each category, an implicit-stage HMM with $K = 7$ regimes is trained to model the sequence of hidden states across layers. Emissions are diagonal Gaussians with parameters $(\mu_k, \sigma_k^2)$, initialized from a pooled sample of step embeddings. Variances are lower-bounded to $10^{-3}$ for stability.

**Expectation–Maximization (EM)** Training proceeds via EM for $n_{\text{iter}} = 10$ iterations: (i) E-step: For each step, given its fixed category, the implicit stage HMM runs forward–backward to compute regime posteriors and sufficient statistics. (ii) M-step: Explicit start and transition probabilities are computed by normalizing the observed category counts from the fixed labels. Implicit stage parameters (regime transitions and Gaussian means/variances) are re-estimated from the posterior-weighted statistics accumulated in the E-step. Average per-step log-likelihood is monitored across iterations.

## B.2 DECODING

**Inputs and Preprocessing Restore** Anchored decoding operates on two inputs: (1) preprocessed records containing step-level hidden states and labels, and (2) a EIRL checkpoint that stores model parameters and preprocessing statistics. The original StandardScaler and PCA are reconstructed by directly restoring their learned attributes, rather than refitting.

**Anchored Decoding** Implicit stage states are decoded with Viterbi under the fixed Explicit stage category path. Anchored decoding constrains the explicit stage EIRL categories using provided labels and returns best_regimes_per_step for each sequence, which is the fine-grained implicit stage regime path assigned by model.

**Consensus Layer Ranges** To summarize regime allocation across layers, we derive *consensus ranges* for each top category:

Table 9: Example.

| Regime | Layers |
|--------|--------|
| 5 | 0–6 |
| 3 | 7–18 |
| 1 | 19–22 |
| 6 | 23–25 |
| 0 | 26–27 |
| 6 | 28 |

1. **Frequency pooling.** For each category $c$, we aggregate all sequences and steps into a frequency tensor $\text{reg\_freq}[c] \in \mathbb{N}^{L \times R}$, where entry $(i, r)$ counts how often regime $r$ is assigned at layer $i$ when the explicit state is $c$.

2. **Dominant regime selection.** At each layer $i$, we assign a single regime by taking the mode of this distribution, $r^* = \arg\max_r \text{reg\_freq}[c]_{i,r}$, yielding a layerwise regime chain for category $c$.

3. **Range consolidation.** Finally, we merge consecutive layers that share the same dominant regime into compact intervals, which we refer to as consensus ranges.

# C STEERING VECTOR CONSTRUCTION

We provide implementation details for the steering vector construction, which derives steering vectors from EIRL transition matrix and hidden states. These vectors capture directional displacements in representation space that distinguish successful from unsuccessful reasoning paths.

---

**Algorithm 1:** Edge-conditioned steering vector construction

---

**Input:** hidden states $\{h_{t,l}\}$, categories $\{c_t\}$, correctness labels $y$, preprocessing $\Phi$, edge set $E$.
**Output:** Steering vectors $v_{a \to b} \in \mathbb{R}^d$.
**for** *each step $t$ in each trajectory* **do**
    $\tilde{h}_t = \Phi(h_{t,l})$, $\tilde{h}_{t+1} = \Phi(h_{t+1,l})$, $d_t = \tilde{h}_{t+1} - \tilde{h}_t$;
    **if** $y = \textbf{\textit{s}}$ **then**
        add $d_t$ to bucket $(a{=}c_t, b{=}c_{t+1})$ for successful trajectory
    **else**
        add $d_t$ to bucket $(a, b)$ for unsuccessful trajectory and to source bucket $a$

**for** *each edge $(a, b) \in E$* **do**
    $\mu_{a \to b}^{\textsf{succ}} = $ mean of successful displacements on $(a, b)$;
    $\mu_{a \to b}^{\textsf{unsucc}} = $ mean of unsuccessful displacements from $a$ excluding $(a, b)$;
    $\Delta_{a \to b} = \text{normalize}(\mu_{a \to b}^{\textsf{succ}} - \mu_{a \to b}^{\textsf{unsucc}})$;
    $v_{a \to b} = \Phi^{-1}(\Delta_{a \to b})$;

---

**Input.** The build script consumes three artifacts: (1) hidden-state records from the base model (with correctness flags and per-step category), (2) explicit stage transition matrices that summarizes category transitions in successful vs. unsuccessful runs, and (3) trained preprocessing parameters that map raw hidden states into the HHMM feature space.

**Preprocessing.** Each hidden vector $h_{t,\ell}$ is standardized using the fitted STANDARDSCALER and projected with PCA to a $d_{\text{pca}}$-dimensional embedding. This yields a representation $v_{t,\ell} \in \mathbb{R}^{d_{\text{pca}}}$ for every reasoning step $t$ and layer $\ell$, aligned with the HHMM training space.

**Edge statistics.** For each semantic edge $(a \to b)$, we compute displacement statistics in the same feature space $\tilde{h}_{t,\ell} = \Phi(h_{t,\ell})$ used in the edge-conditioned vector construction. For every step $t$ with top-level transition $(c_t, c_{t+1}) = (a, b)$, we form the displacement

$$d_t = \tilde{h}_{t+1,\ell} - \tilde{h}_{t,\ell}.$$

We separate displacements by outcome. For successful trajectories, we average all displacements along edge $(a \to b)$:

$$\mu_{a \to b}^{\mathsf{s}} = \frac{1}{N_{a \to b}^{\mathsf{s}}} \sum_{\substack{t:\, y=\mathsf{s}, \\ c_t = a,\, c_{t+1} = b}} d_t.$$

For unsuccessful trajectories, we construct a *source-conditioned baseline* by averaging all displacements that originate at $a$ but do *not* follow the target edge:

$$\mu_{a \to b}^{\mathsf{us}} = \frac{1}{N_a^{\mathsf{us}} - N_{a \to b}^{\mathsf{us}}} \sum_{\substack{t:\, y=\mathsf{us}, \\ c_t = a,\, c_{t+1} \neq b}} d_t.$$

The resulting contrastive edge vector is

$$\Delta_{a \to b} = \mu_{a \to b}^{\mathsf{s}} - \mu_{a \to b}^{\mathsf{us}} \in \mathbb{R}^k,$$

highlighting the displacement components that distinguish successful from unsuccessful progress along this semantic edge. We then normalize $\Delta_{a \to b}$ and map it back to the model's hidden space via $v_{a \to b} = \Phi^{-1}(\Delta_{a \to b}) \in \mathbb{R}^d$ for use in steering.

**Baselines.** In addition to edge-conditioned vectors, we compute an edge-agnostic baseline by averaging embeddings over all steps in successful vs. unsuccessful runs. Their difference defines a edge-agnostic steering direction.

**Soft edge weighting.** We select the top-k semantic edges with the largest difference between successful and unsuccessful transition probabilities, and apply softmax to obtain normalized weights. During steering, we either sample an edge vector according to the weight distribution.

Table 10: Correction rate (↑; fraction of originally incorrect predictions corrected after steering, with standard deviations) and false-token count (↓; token counts of originally incorrect predictions, with standard deviations), average and std over 3 independent runs. Steer at the last layer.

| Model | Dataset | Edge-Agnostic | Soft Steering | | Hard Steering | | Baseline | Soft Steering | | Hard Steering | | #False | $\alpha$ |
|---|---|---|---|---|---|---|---|---|---|---|---|---|---|
| | | Corr. | Corr. | Std | Corr. | Std | Tokens | Tokens | Std | Tokens | Std | | |
| | MATH-500 | 0.2515 | 0.2697 | 0.04 | 0.2212 | 0.07 | 1970 | 1849 | 20.7 | 1845 | 28.4 | 110 | 0.2 |
| Bespoke-Stratos-7B | WebInstruct-Verified | 0.1632 | 0.1875 | 0.02 | 0.1597 | 0.02 | 2938 | 2944 | 44.0 | 2953 | 73.7 | 96 | 0.2 |
| | GPQA-Diamond | 0.1500 | 0.1682 | 0.07 | 0.1773 | 0.06 | 3927 | 3927 | 176.0 | 3781 | 94.0 | 73 | 0.1 |
| | MATH-500 | 0.2126 | 0.2216 | 0.03 | 0.2126 | 0.04 | 1994 | 1994 | 2.0 | 1949 | 13.0 | 111 | 0.1 |
| OpenThinker-7B | WebInstruct-Verified | 0.1448 | 0.1448 | 0.05 | 0.1345 | 0.02 | 3905 | 3804 | 108.0 | 3765 | 60.0 | 97 | 0.2 |
| | GPQA-Diamond | 0.1181 | 0.1266 | 0.03 | 0.1561 | 0.03 | 4700 | 4575 | 128.0 | 4483 | 188.0 | 79 | 0.2 |
| | MATH-500 | 0.1506 | 0.1446 | 0.03 | 0.1627 | 0.05 | 1997 | 1989 | 1.8 | 1982 | 4.5 | 111 | 0.2 |
| Qwen3-1.7B | WebInstruct-Verified | 0.1241 | 0.1448 | 0.01 | 0.1207 | 0.02 | 3793 | 3703 | 56.1 | 3787 | 29.6 | 97 | 0.2 |
| | GPQA-Diamond | 0.0317 | 0.0357 | 0.02 | 0.0516 | 0.04 | 4731 | 4649 | 17.2 | 4604 | 52.4 | 84 | 0.2 |
| | MATH-500 | 0.1562 | 0.1622 | 0.01 | 0.1742 | 0.01 | 1980 | 1980 | 5.1 | 1959 | 6.1 | 111 | 0.1 |
| Llama-3.1-Nemotron-4B-v1.1 | WebInstruct-Verified | 0.0951 | 0.1344 | 0.03 | 0.1279 | 0.04 | 4110 | 4061 | 99.4 | 4099 | 66.1 | 102 | 0.2 |
| | GPQA-Diamond | 0.0565 | 0.0605 | 0.02 | 0.0524 | 0.03 | 4909 | 4793 | 54.7 | 4786 | 72.9 | 83 | 0.2 |
| | **Average** | 0.1379 | 0.1500 | | 0.1459 | | 3413 | 3356 | | 3333 | | 96 | |

## D   EXPLICIT STAGE EIRL TRANSITION ANALYSIS

We characterize the *explicit-stage* semantic dynamics of EIRL by aggregating start distributions and transition matrices across subsets of reasoning trajectories. For any run subset (or *slice*) $\mathcal{D}$—such as all successful or all unsuccessful trajectories—we compute

$$\bar{T} = \text{mean}_{r \in \mathcal{D}} \, T^{(r)} \in \mathbb{R}^{C \times C}, \qquad \bar{S} = \text{mean}_{r \in \mathcal{D}} \, S^{(r)} \in \mathbb{R}^{C},$$

where $T^{(r)}$ and $S^{(r)}$ are the per–run transition matrix and start distribution. To highlight differences between outcomes, we additionally report

$$\Delta T = \bar{T}_{\text{succ}} - \bar{T}_{\text{unsucc}}, \qquad \Delta S = \bar{S}_{\text{succ}} - \bar{S}_{\text{unsucc}}.$$

Results are in Table 17 to Table 43.

## E   IMPLICIT STAGE EIRL TRANSITION ANALYSIS

This section describes the procedure for computing implicit stage divergence between successful and unsuccessful reasoning trajectories using Jaccard similarity of consensus boundary positions.

### E.1   BOUNDARY EXTRACTION

For each reasoning step, Viterbi decoding assigns each layer to one of $K$ regimes. A *regime boundary* occurs at layer $l$ when the regime assignment changes between layers $l - 1$ and $l$. For a model with $L$ layers, we extract boundary positions from each decoded trajectory.

### E.2   WITHIN-MODEL CONSENSUS VOTING

Raw boundary positions vary across datasets and random seeds. To identify stable boundaries, we apply consensus voting within each model. Let $\mathcal{B}_r$ denote the boundary set from run $r$. An interior boundary position $b \notin \{0, B\}$ is retained in the consensus set if:

$$\frac{|\{r : b \in \mathcal{B}_r\}|}{n_{\text{runs}}} \geq \tau$$

where $n_{\text{runs}}$ is the total number of runs and $\tau = 0.25$ is the voting threshold.

### E.3   JACCARD SIMILARITY FOR SUCCESSFUL VS. UNSUCCESSFUL

We quantify structural divergence between successful and unsuccessful trajectories using Jaccard similarity. Let $B_{\text{succ}}^{(c)}$ and $B_{\text{unsucc}}^{(c)}$ denote the consensus boundary positions (excluding fixed endpoints) for category $c$. The Jaccard similarity is:

$$J(c) = \frac{|\mathcal{B}_{\text{succ}}^{(c)} \cap \mathcal{B}_{\text{unsucc}}^{(c)}|}{|\mathcal{B}_{\text{succ}}^{(c)} \cup \mathcal{B}_{\text{unsucc}}^{(c)}|}$$

This metric equals 1 when successful and unsuccessful trajectories place boundaries at identical depths and 0 when they share no common boundaries. We compute $J(c)$ for each model and report the mean, standard deviation, and range across models.

### E.4   CROSS-MODEL SIMILARITY

To enable comparison across models with different layer counts, we normalize absolute boundary positions to percent-depth bins. Each absolute endpoint $e \in [0, L]$ is mapped to a bin $k \in [0, B]$ (with $B = 100$):

$$k = \text{round}\left(\frac{e}{L} \cdot B\right)$$

This normalization allows direct comparison between Qwen-family models and LLaMA-family models. To assess whether models share similar boundary placement, we compute pairwise similarity using histogram overlap. For each model $m$ and category $c$, we construct a histogram

$H_m^{(c)}$ counting boundary occurrences at each percent-depth bin. The normalized histogram is $\hat{H}_m^{(c)} = H_m^{(c)} / \sum_k H_m^{(c)}[k]$. For models $m_1$ and $m_2$, the histogram overlap is:

$$\text{Overlap}(m_1, m_2; c) = \sum_{k=0}^{B} \min \left( \hat{H}_{m_1}^{(c)}[k], \hat{H}_{m_2}^{(c)}[k] \right)$$

This yields 1 when normalized histograms are identical and 0 when they have disjoint support. The overall cross-model similarity averages this overlap across all categories.

## F ABLATION STUDIES

This section examines the sensitivity of our analysis to two key hyperparameters: the number of PCA dimensions(Appendix F.1) and the number of hidden regimes $K$ (Appendix F.2).

### F.1 ABLATION STUDY ON PCA DIMENSIONS

We examine the sensitivity of steering vector construction to the number of PCA dimensions. Table 11 reports steering results across PCA dimensions $d \in \{32, 64, 128\}$.

Table 11: Ablation on PCA dimensions. Bespoke-Stratos-7B on MATH-500 (seed=1, $\alpha$=0.1).

| | Correction Rate | | Tokens | |
|---|---|---|---|---|
| **PCA Dim** | Soft Steering | Hard Steering | Soft Steering | Hard Steering |
| 32 | 0.21 | 0.25 | 1803 | 1845 |
| 64 | 0.21 | 0.25 | 1805 | 1843 |
| 128 | 0.21 | 0.24 | 1807 | 1841 |

Results show that steering performance is stable across PCA dimensions. We use $d = 64$ in our main experiments as it provides a balance between capturing sufficient variance and computational efficiency.

### F.2 ABLATION STUDY ON NUMBER OF REGIMES $K$

We examine the sensitivity of our structural analysis to the number of regimes $K$. Tables 12–14 report family-level boundaries and Jaccard similarity for $K \in \{4, 7, 10\}$, and Table 15 shows steering results using EIRL-identified intermediate transition layers for each $K$.

Table 12: Family-level boundaries with $K = 4$.

| Family | Category | Segments | Jaccard |
|---|---|---|---|
| Qwen | setup | 4 segments: 0–12, 13–20, 21–27, 28 | 0.51 (0.20–0.75) |
| | analysis | 5 segments: 0–13, 14–19, 20–23, 24–27, 28 | 0.40 (0.33–0.50) |
| | verify | 4 segments: 0–21, 22–24, 25–27, 28 | 0.17 (0.09–0.29) |
| | final | 4 segments: 0–19, 20–23, 24–27, 28 | 0.55 (0.33–0.75) |
| LLaMA | setup | 5 segments: 0–12, 13, 14–20, 21–31, 32 | 0.67 |
| | analysis | 5 segments: 0–12, 13–20, 21, 22–31, 32 | 0.50 |
| | verify | 4 segments: 0–12, 13–20, 21–31, 32 | 0.29 |
| | final | 5 segments: 0–12, 13–20, 21, 22–31, 32 | 0.60 |

Table 15: Layer-wise steering results for hard steering. Intermediate layers selected by EIRL boundaries are highlighted: blue for $K$=4 (layers 15, 21, 24), orange for $K$=7 (layers 9, 20, 24, 26), green for $K$=10 (layers 8, 18, 21, 23, 26), with overlapping selections shown in gray.

| Layer | 1 | 2 | 3 | 4 | 5 | 6 | 7 | 8 | 9 | 10 | 11 | 12 | 13 | 14 | 15 | 16 | 17 | 18 | 19 | 20 | 21 | 22 | 23 | 24 | 25 | 26 | 27 | 28 |
|---|---|---|---|---|---|---|---|---|---|---|---|---|---|---|---|---|---|---|---|---|---|---|---|---|---|---|---|---|
| Correction | .24 | .30 | .26 | .24 | .27 | .33 | .24 | .23 | .32 | .27 | .26 | .23 | .35 | .27 | .29 | .26 | .20 | .22 | .21 | .28 | .28 | .28 | .19 | .27 | .19 | .27 | .24 | .21 |
| Tokens | 1826 | 1805 | 1842 | 1850 | 1823 | 1811 | 1801 | 1824 | 1799 | 1800 | 1784 | 1817 | 1775 | 1768 | 1831 | 1815 | 1838 | 1811 | 1836 | 1831 | 1775 | 1820 | 1842 | 1832 | 1859 | 1828 | 1832 | 1805 |

Table 13: Family-level boundaries with $K = 7$.

| Family | Category | Segments | Jaccard |
|---|---|---|---|
| Qwen | setup | 11 segments: 0–10, 11–18, 19, 20, 21, 22, 23, 24, 25, 26–27, 28 | 0.71 (0.64–0.83) |
| | analysis | 10 segments: 0–7, 8–18, 19, 20–21, 22, 23, 24, 25, 26–27, 28 | 0.66 (0.50–0.80) |
| | verify | 11 segments: 0–17, 18, 19, 20, 21, 22, 23, 24, 25, 26–27, 28 | 0.56 (0.46–0.64) |
| | final | 9 segments: 0–7, 8–17, 18–19, 20, 21, 22–24, 25, 26–27, 28 | 0.56 (0.46–0.73) |
| LLaMA | setup | 11 segments: 0–9, 10, 11–12, 13–16, 17–18, 19, 20–22, 23–26, 27, 28–31, 32 | 0.56 |
| | analysis | 9 segments: 0–10, 11, 12–17, 18, 19–25, 26, 27–28, 29–31, 32 | 0.56 |
| | verify | 7 segments: 0–12, 13–16, 17, 18–20, 21–23, 24–31, 32 | 0.38 |
| | final | 11 segments: 0–9, 10, 11, 12–16, 17, 18–22, 23–25, 26, 27, 28–31, 32 | 0.17 |

Table 14: Family-level boundaries with $K = 10$.

| Family | Category | Segments | Jaccard |
|---|---|---|---|
| Qwen | setup | 14 segments: 0–7, 8–10, 11–16, 17, 18, 19, 20, 21, 22, 23, 24, 25, 26–27, 28 | 0.49 (0.42–0.54) |
| | analysis | 14 segments: 0–6, 7, 8–16, 17, 18, 19, 20, 21, 22, 23, 24, 25, 26–27, 28 | 0.72 (0.64–0.83) |
| | verify | 13 segments: 0–7, 8–10, 11–16, 17–18, 19, 20, 21, 22, 23, 24, 25, 26–27, 28 | 0.68 (0.54–0.83) |
| | final | 11 segments: 0–7, 8–16, 17–18, 19, 20, 21, 22–24, 25, 26–27, 28 | 0.76 (0.64–0.83) |
| LLaMA | setup | 11 segments: 0–7, 8–10, 11, 12–15, 16–17, 18, 19–20, 21–26, 27, 28–31, 32 | 0.50 |
| | analysis | 11 segments: 0–9, 10, 11–15, 16, 17, 18–21, 22–25, 26–27, 28, 29–31, 32 | 0.50 |
| | verify | 10 segments: 0–9, 10, 11–16, 17, 18–20, 21, 22–26, 27, 28–31, 32 | 0.58 |
| | final | 13 segments: 0–9, 10, 11–16, 17, 18–20, 21–22, 23–24, 25, 26, 27, 28–29, 30–31, 32 | 0.40 |

The choice of $K$ critically affects segmentation granularity. With $K = 4$, the model produces overly coarse boundaries, potentially obscuring meaningful structural differences. With $K = 10$, excessive fragmentation occurs: categories exhibit boundaries at nearly every layer in the upper half of the network, making it difficult to distinguish true transition signal. The resulting high Jaccard values likely reflect this over-segmentation rather than genuine structural similarity. We select $K = 7$ as it balances interpretability with discriminative power: segments are granular enough to capture meaningful regime transitions while avoiding the noise introduced by over-fragmentation. Our main conclusions are robust across different $K$ values. The characteristic "delayed first boundary" pattern in *verify* persists across all three settings, and *verify* consistently exhibits relatively low boundary overlap between correct and incorrect trajectories at $K = 4$ and $K = 7$. For steering effectiveness, Table 15 shows that EIRL-selected intermediate layers identify local peaks in correction rates for both $K = 4$ and $K = 7$, demonstrating that our structural analysis reliably pinpoints layers where steering interventions are effective.

## G  BOOTSTRAP ANALYSIS OF TRANSITION-DIFFERENCE STATISTICAL SIGNIFICANCE

To assess whether the observed differences between the explicit stage transition dynamics of successful and unsuccessful trajectories are *stable* across model/dataset/seed, we apply a bootstrap analysis over all runs. Each run $r$ yields one EIRL-estimated transition matrix $A_{i,j}^{exp(r)}(succ)$ and $A_{i.j}^{exp(r)}(unsucc)$. Let $R$ denote the total number of runs. For each bootstrap replicate $b$, we resample the run indices with replacement, $\mathcal{I}^{(b)} \subseteq \{1, \ldots, R\}$, $\quad |\mathcal{I}^{(b)}| = R$, and compute the mean transition-difference matrix

$$\Delta_{i,j}^{(b)} = \frac{1}{R} \sum_{r \in \mathcal{I}^{(b)}} A_{i,j}^{exp(r)}(succ) - \frac{1}{R} \sum_{r \in \mathcal{I}^{(b)}} A_{i.j}^{exp(r)}(unsucc).$$

Repeating this procedure (we use $B = 1000$ resamples) yields an empirical sampling distribution for each edge difference $\Delta_{i,j}$, from which we obtain percentile-based confidence intervals.

This bootstrap quantifies the *significance* of observed differences across runs. A narrow bootstrap interval indicates that the difference is consistently reproduced across models, datasets, and seeds. Table 16 reports the 95% confidence intervals for $\Delta_{i,j}$ for all significant edges. Several edges—notably `final_answer→final_answer`, `analysis_and_computation→final_answer`, and `uncertainty_and_verification→final_answer`—show consistently positive intervals, indicating that successful trajectories are reliably more likely to transition toward

final_answer. Edges such as setup_and_retrieval→analysis_and_computation and analysis_and_computation→uncertainty_and_verification likewise show stable negative or positive differences. In contrast, edges whose intervals include zero (e.g., analysis_and_computation→setup_and_retrieval) exhibit substantial run-to-run variability. We also test AVA loops and its 95% CI is [-0.013, -0.005] (unsuccessful > successful).

Table 16: 95% bootstrap confidence intervals for $\Delta = A(succ) - A(unsucc)$.

| Edge | 95% CI | Interpretation |
|---|---|---|
| final_answer→final_answer | [0.008, 0.075] | successful > unsuccessful |
| final_answer→analysis_and_computation | [-0.071, -0.024] | unsuccessful > successful |
| setup→setup | [-0.044, -0.012] | unsuccessful > successful |
| setup→analysis_and_computation | [0.025, 0.058] | successful > unsuccessful |
| analysis_and_computation→final_answer | [0.032, 0.058] | successful > unsuccessful |
| analysis_and_computation→analysis_and_computation | [-0.033, -0.007] | unsuccessful > successful |
| analysis_and_computation→verification | [-0.031, -0.009] | unsuccessful > successful |
| verification→final_answer | [0.044, 0.088] | successful > unsuccessful |
| verification→verification | [-0.065, -0.022] | unsuccessful > successful |

# H QUALITATIVE EXAMPLE OF STEERING EFFECT ON STATE TRANSITIONS

To illustrate how steering affects reasoning dynamics, we present a qualitative example by using the steering vector that encourage *uncertainty_and_verification → final_answer*. Consider the following problem from MATH-500:

**Question:**

The data in the table below shows the percent of bus riders in a survey of Central H.S. students; 300 males and 300 females in each grade were surveyed. For which grade is the number of its male bus riders closest to $135\%$ of the number of its female bus riders?

| Grade | Males | Females |
|---|---|---|
| $9^{th}$ grade | 41.1% | 39.4% |
| $10^{th}$ grade | 34.4% | 33.1% |
| $11^{th}$ grade | 20.6% | 13.8% |
| $12^{th}$ grade | 11.6% | 8.6% |

**Answer:** 12

Below we show each *uncertainty_and_verification* step and its subsequent transition. The following responses are from Bespoke-Stratos-7B with seed=1 and $\alpha$ =0.2.

**Baseline Response** Baseline Response fail to get correct answer in 2000 tokens.

*Hmm, closer than 9th grade (12.09% vs 10.285%?), but maybe not. Let's check others.* (UNCERTAINTY_AND_VERIFICATION)

*11th grade.* (SETUP_AND_RETRIEVAL) . . .

*Wow, that's much closer. Only 1.97% difference. That seems really close.* (UNCERTAINTY_AND_VERIFICATION)

*Let me check 12th grade just to be thorough.* (UNCERTAINTY_AND_VERIFICATION) . . .

*That's even smaller! Wait, only 0.01% difference. How is that possible?* (UNCERTAINTY_AND_VERIFICATION)

*Wait, let me verify that calculation again because 11.6% vs 11.61% is only 0.01% apart. That would mean they are practically the same. But how does that happen?* (UNCERTAINTY_AND_VERIFICATION) . . .

*But wait, the question says "closest to," so 12th grade would be the answer. However, before finalizing, let me check all grades again because sometimes there might be a trick.* (UNCERTAINTY_AND_VERIFICATION)

*Looking back at the calculations.* (ANALYSIS_AND_COMPUTATION) . . .

*But wait, hold on. Let me check if my calculation for 12th grade is correct. 8.6% multiplied by 1.35.* (UNCERTAINTY_AND_VERIFICATION)

*First, $8.6 \times 1 = 8.6$.* (ANALYSIS_AND_COMPUTATION) . . .

*But wait, let me check once more to ensure no miscalculations. Maybe I misread the percentages.* (UNCERTAINTY_AND_VERIFICATION)

*Looking at the table.* (SETUP_AND_RETRIEVAL) . . .

**Steered Response** Steered Response get correct answer by generating 1846 tokens.

After applying the steering vector, the model transitions more decisively from *uncertainty_and_verification* directly to *final_answer*:

*Wait a minute, the difference here is only 0.07, which is much smaller than the others. So 12th grade has a very small difference. Let me double-check my calculations to make sure I didn't make a mistake.* (UNCERTAINTY_AND_VERIFICATION)

*For 12th grade:* (SETUP_AND_RETRIEVAL)

. . .

*But wait, let me confirm once again. If the difference is 0.07, that means 34.8 vs. 34.73. So, 34.8 is 0.07 more than 34.73. That's a very tiny difference, almost negligible. Therefore, the 12th grade is the answer because the number of male bus riders is closest to 135% of female bus riders there.* (UNCERTAINTY_AND_VERIFICATION)

*So yes, 12th grade is the answer.* (FINAL_ANSWER)

**Analysis** In the baseline response, the model enters *uncertainty_and_verification* 8 times, with only 1 transition (12.5%) leading directly to *final_answer*—the remaining transitions loop back to *uncertainty_and_verification* (2 times), *analysis_and_computation* (3 times), or *setup_and_retrieval* (2 times). After steering, the model enters *uncertainty_and_verification* only 2 times, with 1 transition leading directly to *final_answer*, demonstrating that steering reduces unproductive verification loops.

## I    USE OF LARGE LANGUAGE MODELS.

Large language models (LLMs) were used solely as assistive tools for proofreading and improving clarity of writing.

Table 17: Mean top–level transition matrices ($\bar{T}$). Rows are sources; columns are destinations.

| | final_answer | setup_and_retrieval | analysis_and_computation | uncertainty_and_verification | unknown |
|---|---|---|---|---|---|
| *ALL* | | | | | |
| final_answer | 0.497 | 0.111 | 0.272 | 0.117 | 0.001 |
| setup_and_retrieval | 0.245 | 0.323 | 0.355 | 0.076 | 0.002 |
| analysis_and_computation | 0.245 | 0.132 | 0.505 | 0.116 | 0.002 |
| uncertainty_and_verification | 0.202 | 0.168 | 0.362 | 0.267 | 0.002 |
| unknown | 0.222 | 0.313 | 0.381 | 0.063 | 0.021 |
| *CORRECT* | | | | | |
| final_answer | 0.515 | 0.110 | 0.252 | 0.123 | 0.001 |
| setup_and_retrieval | 0.243 | 0.300 | 0.387 | 0.068 | 0.002 |
| analysis_and_computation | 0.276 | 0.128 | 0.493 | 0.101 | 0.002 |
| uncertainty_and_verification | 0.246 | 0.158 | 0.359 | 0.236 | 0.001 |
| unknown | 0.244 | 0.290 | 0.325 | 0.086 | 0.054 |
| *INCORRECT* | | | | | |
| final_answer | 0.471 | 0.112 | 0.298 | 0.118 | 0.002 |
| setup_and_retrieval | 0.245 | 0.328 | 0.346 | 0.080 | 0.002 |
| analysis_and_computation | 0.231 | 0.133 | 0.512 | 0.121 | 0.002 |
| uncertainty_and_verification | 0.181 | 0.171 | 0.367 | 0.279 | 0.002 |
| unknown | 0.213 | 0.313 | 0.379 | 0.068 | 0.028 |

Table 18: Difference matrix $\Delta T = \bar{T}_{\text{correct}} - \bar{T}_{\text{incorrect}}$. Positive values indicate transitions that are *more likely* in correct runs.

| | final_answer | setup_and_retrieval | analysis_and_computation | uncertainty_and_verification | unknown |
|---|---|---|---|---|---|
| final_answer | **+0.044** | -0.002 | -0.047 | +0.004 | -0.000 |
| setup_and_retrieval | -0.002 | -0.028 | **+0.042** | -0.012 | +0.001 |
| analysis_and_computation | **+0.045** | -0.006 | -0.020 | -0.020 | -0.000 |
| uncertainty_and_verification | **+0.065** | -0.013 | -0.008 | -0.043 | -0.001 |
| unknown | +0.032 | -0.022 | -0.053 | +0.018 | +0.026 |

Table 19: Mean start distributions ($\bar{S}$) and difference $\Delta S = \bar{S}_{\text{correct}} - \bar{S}_{\text{incorrect}}$.

| | **ALL** | **CORRECT** | **INCORRECT** |
|---|---|---|---|
| final_answer | 0.084 | 0.087 | 0.082 |
| setup_and_retrieval | 0.484 | 0.458 | 0.490 |
| analysis_and_computation | 0.287 | 0.321 | 0.268 |
| uncertainty_and_verification | 0.015 | 0.015 | 0.016 |
| unknown | 0.131 | 0.119 | 0.144 |
| *Difference* $\Delta S$ (Correct − Incorrect): [ +0.005, -0.032, +0.052, -0.001, -0.024 ] | | | |

Table 20: AIME-2024 — Mean top–level transition matrices ($\bar{T}$). Rows are sources; columns are destinations.

| | final_answer | setup_and_retrieval | analysis_and_computation | uncertainty_and_verification | unknown |
|---|---|---|---|---|---|
| *ALL* | | | | | |
| final_answer | 0.481 | 0.112 | 0.324 | 0.083 | 0.001 |
| setup_and_retrieval | 0.236 | 0.327 | 0.384 | 0.051 | 0.002 |
| analysis_and_computation | 0.239 | 0.130 | 0.535 | 0.094 | 0.002 |
| uncertainty_and_verification | 0.191 | 0.173 | 0.386 | 0.248 | 0.001 |
| unknown | 0.161 | 0.399 | 0.356 | 0.065 | 0.020 |
| *CORRECT* | | | | | |
| final_answer | 0.455 | 0.125 | 0.322 | 0.097 | 0.001 |
| setup_and_retrieval | 0.213 | 0.294 | 0.459 | 0.032 | 0.003 |
| analysis_and_computation | 0.261 | 0.144 | 0.520 | 0.073 | 0.002 |
| uncertainty_and_verification | 0.207 | 0.182 | 0.410 | 0.202 | 0.000 |
| unknown | 0.182 | 0.293 | 0.356 | 0.113 | 0.056 |
| *INCORRECT* | | | | | |
| final_answer | 0.478 | 0.109 | 0.331 | 0.081 | 0.001 |
| setup_and_retrieval | 0.247 | 0.336 | 0.361 | 0.056 | 0.001 |
| analysis_and_computation | 0.231 | 0.125 | 0.545 | 0.098 | 0.002 |
| uncertainty_and_verification | 0.189 | 0.172 | 0.385 | 0.254 | 0.001 |
| unknown | 0.164 | 0.417 | 0.345 | 0.057 | 0.017 |

Table 21: AIME-2024 — Difference matrix $\Delta T = \bar{T}_{\text{correct}} - \bar{T}_{\text{incorrect}}$. Positive values are more likely in correct runs.

| | final_answer | setup_and_retrieval | analysis_and_computation | uncertainty_and_verification | unknown |
|---|---|---|---|---|---|
| from_final | -0.024 | +0.016 | -0.008 | +0.016 | -0.000 |
| from_setup | -0.034 | -0.042 | +0.098 | -0.024 | +0.002 |
| from_analysis | +0.030 | +0.020 | -0.025 | -0.025 | +0.001 |
| from_verify | +0.018 | +0.010 | +0.025 | -0.052 | -0.001 |
| from_unknown | +0.018 | -0.124 | +0.011 | +0.055 | +0.040 |

Table 22: AIME-2024 — Mean start distributions ($\bar{S}$) and $\Delta S = \bar{S}_{\text{correct}} - \bar{S}_{\text{incorrect}}$.

| | ALL | CORRECT | INCORRECT |
|---|---|---|---|
| final_answer | 0.100 | 0.113 | 0.081 |
| setup_and_retrieval | 0.489 | 0.484 | 0.471 |
| analysis_and_computation | 0.217 | 0.250 | 0.222 |
| uncertainty_and_verification | 0.000 | 0.000 | 0.000 |
| unknown | 0.194 | 0.154 | 0.226 |

*Difference $\Delta S$ (Correct − Incorrect): [ +0.032, +0.013, +0.028, +0.000, -0.072 ]*

Table 23: GPQA-Diamond — Mean top–level transition matrices ($\bar{T}$). Rows are sources; columns are destinations.

| | final_answer | setup_and_retrieval | analysis_and_computation | uncertainty_and_verification | unknown |
|---|---|---|---|---|---|
| *ALL* | | | | | |
| from_final | 0.499 | 0.135 | 0.225 | 0.139 | 0.002 |
| from_setup | 0.232 | 0.365 | 0.270 | 0.132 | 0.001 |
| from_analysis | 0.210 | 0.156 | 0.440 | 0.192 | 0.003 |
| from_verify | 0.191 | 0.171 | 0.301 | 0.335 | 0.002 |
| from_unknown | 0.351 | 0.281 | 0.270 | 0.062 | 0.036 |
| *CORRECT* | | | | | |
| from_final | 0.529 | 0.119 | 0.207 | 0.143 | 0.002 |
| from_setup | 0.231 | 0.328 | 0.310 | 0.128 | 0.003 |
| from_analysis | 0.253 | 0.143 | 0.441 | 0.159 | 0.004 |
| from_verify | 0.264 | 0.135 | 0.266 | 0.333 | 0.002 |
| from_unknown | 0.321 | 0.334 | 0.186 | 0.093 | 0.067 |
| *INCORRECT* | | | | | |
| from_final | 0.475 | 0.144 | 0.237 | 0.141 | 0.003 |
| from_setup | 0.232 | 0.369 | 0.265 | 0.133 | 0.001 |
| from_analysis | 0.203 | 0.158 | 0.438 | 0.198 | 0.003 |
| from_verify | 0.176 | 0.176 | 0.308 | 0.337 | 0.002 |
| from_unknown | 0.328 | 0.253 | 0.291 | 0.076 | 0.052 |

Table 24: GPQA-Diamond — Difference matrix $\Delta T = \bar{T}_{\text{correct}} - \bar{T}_{\text{incorrect}}$. Positive values are more likely in correct runs.

| | final_answer | setup_and_retrieval | analysis_and_computation | uncertainty_and_verification | unknown |
|---|---|---|---|---|---|
| from_final | +0.053 | -0.025 | -0.030 | +0.002 | -0.000 |
| from_setup | -0.001 | -0.042 | +0.045 | -0.004 | +0.001 |
| from_analysis | +0.050 | -0.015 | +0.002 | -0.039 | +0.001 |
| from_verify | +0.088 | -0.041 | -0.043 | -0.004 | -0.000 |
| from_unknown | -0.007 | +0.081 | -0.105 | +0.017 | +0.014 |

Table 25: GPQA-Diamond — Mean start distributions ($\bar{S}$) and $\Delta S = \bar{S}_{\text{correct}} - \bar{S}_{\text{incorrect}}$.

| | ALL | CORRECT | INCORRECT |
|---|---|---|---|
| final_answer | 0.090 | 0.108 | 0.087 |
| setup_and_retrieval | 0.486 | 0.412 | 0.505 |
| analysis_and_computation | 0.240 | 0.272 | 0.229 |
| uncertainty_and_verification | 0.023 | 0.024 | 0.021 |
| unknown | 0.162 | 0.184 | 0.157 |

*Difference $\Delta S$ (Correct − Incorrect): [ +0.021, -0.094, +0.043, +0.003, +0.027 ]*

Table 26: MATH-500 — Mean top–level transition matrices ($\bar{T}$). Rows are sources; columns are destinations.

| | final_answer | setup_and_retrieval | analysis_and_computation | uncertainty_and_verification | unknown |
|---|---|---|---|---|---|
| *ALL* | | | | | |
| from_final | 0.516 | 0.117 | 0.262 | 0.104 | 0.001 |
| from_setup | 0.228 | 0.311 | 0.418 | 0.041 | 0.002 |
| from_analysis | 0.243 | 0.146 | 0.534 | 0.074 | 0.003 |
| from_verify | 0.211 | 0.192 | 0.413 | 0.182 | 0.001 |
| from_unknown | 0.242 | 0.306 | 0.391 | 0.036 | 0.024 |
| *CORRECT* | | | | | |
| from_final | 0.563 | 0.118 | 0.213 | 0.104 | 0.001 |
| from_setup | 0.242 | 0.301 | 0.425 | 0.030 | 0.002 |
| from_analysis | 0.285 | 0.137 | 0.510 | 0.065 | 0.002 |
| from_verify | 0.283 | 0.182 | 0.401 | 0.133 | 0.002 |
| from_unknown | 0.211 | 0.296 | 0.352 | 0.081 | 0.060 |
| *INCORRECT* | | | | | |
| from_final | 0.443 | 0.110 | 0.339 | 0.106 | 0.002 |
| from_setup | 0.214 | 0.319 | 0.412 | 0.053 | 0.002 |
| from_analysis | 0.207 | 0.154 | 0.554 | 0.083 | 0.003 |
| from_verify | 0.148 | 0.201 | 0.433 | 0.216 | 0.001 |
| from_unknown | 0.239 | 0.282 | 0.404 | 0.039 | 0.036 |

Table 27: MATH-500 — Difference matrix $\Delta T = \bar{T}_{\text{correct}} - \bar{T}_{\text{incorrect}}$. Positive values are more likely in correct runs.

| | final_answer | setup_and_retrieval | analysis_and_computation | uncertainty_and_verification | unknown |
|---|---|---|---|---|---|
| from_final | +0.120 | +0.008 | -0.126 | -0.002 | -0.001 |
| from_setup | +0.028 | -0.017 | +0.013 | -0.023 | -0.001 |
| from_analysis | +0.078 | -0.016 | -0.044 | -0.018 | -0.000 |
| from_verify | +0.135 | -0.019 | -0.032 | -0.084 | +0.000 |
| from_unknown | -0.028 | +0.013 | -0.052 | +0.042 | +0.024 |

Table 28: MATH-500 — Mean start distributions ($\bar{S}$) and $\Delta S = \bar{S}_{\text{correct}} - \bar{S}_{\text{incorrect}}$.

| | ALL | CORRECT | INCORRECT |
|---|---|---|---|
| final_answer | 0.076 | 0.070 | 0.086 |
| setup_and_retrieval | 0.452 | 0.439 | 0.470 |
| analysis_and_computation | 0.370 | 0.403 | 0.322 |
| uncertainty_and_verification | 0.014 | 0.011 | 0.019 |
| unknown | 0.088 | 0.077 | 0.103 |

*Difference $\Delta S$ (Correct − Incorrect): [ -0.016, -0.031, +0.081, -0.008, -0.026 ]*

Table 29: WebInstruct-Verified — Mean top–level transition matrices ($\bar{T}$). Rows are sources; columns are destinations.

| | final_answer | setup_and_retrieval | analysis_and_computation | uncertainty_and_verification | unknown |
|---|---|---|---|---|---|
| *ALL* | | | | | |
| from_final | 0.494 | 0.081 | 0.279 | 0.144 | 0.001 |
| from_setup | 0.285 | 0.287 | 0.348 | 0.078 | 0.002 |
| from_analysis | 0.289 | 0.094 | 0.510 | 0.105 | 0.002 |
| from_verify | 0.215 | 0.134 | 0.346 | 0.303 | 0.002 |
| from_unknown | 0.135 | 0.265 | 0.505 | 0.089 | 0.005 |
| *CORRECT* | | | | | |
| from_final | 0.513 | 0.076 | 0.264 | 0.146 | 0.001 |
| from_setup | 0.284 | 0.276 | 0.356 | 0.081 | 0.003 |
| from_analysis | 0.306 | 0.086 | 0.500 | 0.107 | 0.001 |
| from_verify | 0.229 | 0.132 | 0.360 | 0.278 | 0.001 |
| from_unknown | 0.263 | 0.240 | 0.407 | 0.057 | 0.033 |
| *INCORRECT* | | | | | |
| from_final | 0.485 | 0.083 | 0.286 | 0.144 | 0.001 |
| from_setup | 0.285 | 0.289 | 0.346 | 0.078 | 0.003 |
| from_analysis | 0.282 | 0.097 | 0.513 | 0.105 | 0.003 |
| from_verify | 0.209 | 0.135 | 0.343 | 0.310 | 0.003 |
| from_unknown | 0.120 | 0.299 | 0.474 | 0.100 | 0.007 |

Table 30: WebInstruct-Verified — Difference matrix $\Delta T = \bar{T}_{\text{correct}} - \bar{T}_{\text{incorrect}}$. Positive values are more likely in correct runs.

| | final_answer | setup_and_retrieval | analysis_and_computation | uncertainty_and_verification | unknown |
|---|---|---|---|---|---|
| from_final | +0.027 | -0.008 | -0.022 | +0.002 | +0.000 |
| from_setup | -0.001 | -0.013 | +0.010 | +0.003 | +0.000 |
| from_analysis | +0.023 | -0.011 | -0.013 | +0.002 | -0.001 |
| from_verify | +0.020 | -0.003 | +0.017 | -0.032 | -0.002 |
| from_unknown | +0.143 | -0.060 | -0.068 | -0.043 | +0.027 |

Table 31: WebInstruct-Verified — Mean start distributions ($\bar{S}$) and $\Delta S = \bar{S}_{\text{correct}} - \bar{S}_{\text{incorrect}}$.

| | ALL | CORRECT | INCORRECT |
|---|---|---|---|
| final_answer | 0.068 | 0.057 | 0.073 |
| setup_and_retrieval | 0.508 | 0.498 | 0.515 |
| analysis_and_computation | 0.320 | 0.357 | 0.299 |
| uncertainty_and_verification | 0.025 | 0.026 | 0.024 |
| unknown | 0.079 | 0.061 | 0.088 |

*Difference $\Delta S$ (Correct − Incorrect):* [ -0.016, -0.017, +0.057, +0.002, -0.027 ]

Table 32: Nemotron — Mean top–level transition matrices ($\bar{T}$). Rows are sources; columns are destinations.

| | final_answer | setup_and_retrieval | analysis_and_computation | uncertainty_and_verification | unknown |
|---|---|---|---|---|---|
| *ALL* | | | | | |
| final_answer | 0.638 | 0.087 | 0.238 | 0.033 | 0.005 |
| setup_and_retrieval | 0.652 | 0.133 | 0.189 | 0.021 | 0.005 |
| analysis_and_computation | 0.505 | 0.089 | 0.360 | 0.039 | 0.008 |
| uncertainty_and_verification | 0.511 | 0.095 | 0.309 | 0.079 | 0.007 |
| unknown | 0.550 | 0.096 | 0.284 | 0.053 | 0.018 |
| *CORRECT* | | | | | |
| final_answer | 0.659 | 0.074 | 0.216 | 0.047 | 0.005 |
| setup_and_retrieval | 0.634 | 0.116 | 0.208 | 0.033 | 0.008 |
| analysis_and_computation | 0.528 | 0.085 | 0.330 | 0.048 | 0.008 |
| uncertainty_and_verification | 0.534 | 0.079 | 0.290 | 0.093 | 0.004 |
| unknown | 0.634 | 0.104 | 0.195 | 0.050 | 0.017 |
| *INCORRECT* | | | | | |
| final_answer | 0.626 | 0.091 | 0.250 | 0.029 | 0.005 |
| setup_and_retrieval | 0.655 | 0.137 | 0.184 | 0.019 | 0.004 |
| analysis_and_computation | 0.493 | 0.091 | 0.372 | 0.036 | 0.008 |
| uncertainty_and_verification | 0.491 | 0.100 | 0.328 | 0.074 | 0.007 |
| unknown | 0.521 | 0.089 | 0.334 | 0.044 | 0.012 |

Table 33: Nemotron — Difference matrix $\Delta T = \bar{T}_{\text{correct}} - \bar{T}_{\text{incorrect}}$. Positive values are more likely in correct runs.

| | final_answer | setup_and_retrieval | analysis_and_computation | uncertainty_and_verification | unknown |
|---|---|---|---|---|---|
| final_answer | **+0.033** | -0.017 | -0.034 | +0.018 | +0.000 |
| setup_and_retrieval | -0.021 | -0.021 | +0.024 | +0.014 | +0.004 |
| analysis_and_computation | +0.035 | -0.006 | -0.042 | +0.013 | -0.000 |
| uncertainty_and_verification | +0.043 | -0.020 | -0.038 | +0.019 | -0.003 |
| unknown | +0.113 | +0.015 | -0.139 | +0.006 | +0.005 |

Table 34: Nemotron — Mean start distributions ($\bar{S}$) and $\Delta S = \bar{S}_{\text{correct}} - \bar{S}_{\text{incorrect}}$.

| | ALL | CORRECT | INCORRECT |
|---|---|---|---|
| final_answer | 0.331 | 0.345 | 0.324 |
| setup_and_retrieval | 0.100 | 0.095 | 0.080 |
| analysis_and_computation | 0.527 | 0.524 | 0.554 |
| uncertainty_and_verification | 0.031 | 0.024 | 0.035 |
| unknown | 0.011 | 0.011 | 0.008 |

*Difference* $\Delta S$ (Correct − Incorrect): [ +0.021, +0.015, -0.029, -0.010, +0.003 ]

Table 35: Openthinker — Mean top–level transition matrices ($\bar{T}$). Rows are sources; columns are destinations.

| | final_answer | setup_and_retrieval | analysis_and_computation | uncertainty_and_verification | unknown |
|---|---|---|---|---|---|
| *ALL* | | | | | |
| final_answer | 0.530 | 0.105 | 0.228 | 0.137 | 0.001 |
| setup_and_retrieval | 0.100 | 0.388 | 0.419 | 0.092 | 0.000 |
| analysis_and_computation | 0.164 | 0.122 | 0.567 | 0.147 | 0.000 |
| uncertainty_and_verification | 0.119 | 0.148 | 0.384 | 0.350 | 0.000 |
| unknown | 0.100 | 0.217 | 0.575 | 0.058 | 0.050 |
| *CORRECT* | | | | | |
| final_answer | 0.545 | 0.113 | 0.188 | 0.153 | 0.000 |
| setup_and_retrieval | 0.090 | 0.364 | 0.465 | 0.080 | 0.000 |
| analysis_and_computation | 0.204 | 0.108 | 0.555 | 0.133 | 0.001 |
| uncertainty_and_verification | 0.203 | 0.134 | 0.365 | 0.299 | 0.000 |
| unknown | 0.117 | 0.117 | 0.533 | 0.117 | 0.117 |
| *INCORRECT* | | | | | |
| final_answer | 0.487 | 0.102 | 0.273 | 0.136 | 0.001 |
| setup_and_retrieval | 0.105 | 0.392 | 0.406 | 0.097 | 0.000 |
| analysis_and_computation | 0.149 | 0.125 | 0.574 | 0.152 | 0.000 |
| uncertainty_and_verification | 0.083 | 0.154 | 0.394 | 0.369 | 0.000 |
| unknown | 0.100 | 0.262 | 0.529 | 0.058 | 0.050 |

Table 36: Openthinker — Difference matrix $\Delta T = \bar{T}_{\text{correct}} - \bar{T}_{\text{incorrect}}$. Positive values are more likely in correct runs.

| | final_answer | setup_and_retrieval | analysis_and_computation | uncertainty_and_verification | unknown |
|---|---|---|---|---|---|
| final_answer | **+0.058** | +0.012 | -0.085 | +0.017 | -0.001 |
| setup_and_retrieval | -0.014 | -0.028 | +0.059 | -0.017 | -0.000 |
| analysis_and_computation | +0.055 | -0.017 | -0.019 | -0.019 | +0.000 |
| uncertainty_and_verification | +0.120 | -0.020 | -0.029 | -0.070 | +0.000 |
| unknown | +0.017 | -0.146 | +0.004 | +0.058 | +0.067 |

Table 37: Openthinker — Mean start distributions ($\bar{S}$) and $\Delta S = \bar{S}_{\text{correct}} - \bar{S}_{\text{incorrect}}$.

| | ALL | CORRECT | INCORRECT |
|---|---|---|---|
| final_answer | 0.000 | 0.000 | 0.000 |
| setup_and_retrieval | 0.967 | 0.951 | 0.970 |
| analysis_and_computation | 0.030 | 0.043 | 0.028 |
| uncertainty_and_verification | 0.003 | 0.005 | 0.002 |
| unknown | 0.000 | 0.000 | 0.000 |

*Difference* $\Delta S$ (Correct − Incorrect): [ 0.000, -0.019, +0.016, +0.003, 0.000 ]

Table 38: Qwen — Mean top–level transition matrices ($\bar{T}$). Rows are sources; columns are destinations.

| | final_answer | setup_and_retrieval | analysis_and_computation | uncertainty_and_verification | unknown |
|---|---|---|---|---|---|
| *ALL* | | | | | |
| final_answer | 0.367 | 0.155 | 0.357 | 0.121 | 0.000 |
| setup_and_retrieval | 0.098 | 0.392 | 0.412 | 0.098 | 0.000 |
| analysis_and_computation | 0.142 | 0.183 | 0.549 | 0.126 | 0.000 |
| uncertainty_and_verification | 0.068 | 0.262 | 0.405 | 0.265 | 0.000 |
| unknown | 0.009 | 0.599 | 0.312 | 0.080 | 0.000 |
| *CORRECT* | | | | | |
| final_answer | 0.399 | 0.154 | 0.345 | 0.101 | 0.000 |
| setup_and_retrieval | 0.121 | 0.365 | 0.440 | 0.074 | 0.000 |
| analysis_and_computation | 0.179 | 0.180 | 0.550 | 0.091 | 0.000 |
| uncertainty_and_verification | 0.097 | 0.239 | 0.422 | 0.242 | 0.000 |
| unknown | 0.004 | 0.580 | 0.322 | 0.094 | 0.000 |
| *INCORRECT* | | | | | |
| final_answer | 0.338 | 0.149 | 0.383 | 0.130 | 0.000 |
| setup_and_retrieval | 0.090 | 0.397 | 0.403 | 0.110 | 0.000 |
| analysis_and_computation | 0.124 | 0.185 | 0.554 | 0.137 | 0.000 |
| uncertainty_and_verification | 0.058 | 0.269 | 0.394 | 0.279 | 0.000 |
| unknown | 0.009 | 0.603 | 0.313 | 0.075 | 0.000 |

Table 39: Qwen — Difference matrix $\Delta T = \bar{T}_{\text{correct}} - \bar{T}_{\text{incorrect}}$. Positive values are more likely in correct runs.

| | final_answer | setup_and_retrieval | analysis_and_computation | uncertainty_and_verification | unknown |
|---|---|---|---|---|---|
| final_answer | **+0.062** | +0.005 | -0.038 | -0.029 | 0.000 |
| setup_and_retrieval | +0.031 | -0.032 | +0.037 | -0.036 | 0.000 |
| analysis_and_computation | +0.055 | -0.005 | -0.004 | -0.046 | -0.000 |
| uncertainty_and_verification | +0.040 | -0.030 | +0.027 | -0.037 | 0.000 |
| unknown | -0.005 | -0.023 | +0.009 | +0.019 | 0.000 |

Table 40: Qwen — Mean start distributions ($\bar{S}$) and $\Delta S = \bar{S}_{\text{correct}} - \bar{S}_{\text{incorrect}}$.

| | **ALL** | **CORRECT** | **INCORRECT** |
|---|---|---|---|
| final_answer | 0.004 | 0.002 | 0.003 |
| setup_and_retrieval | 0.047 | 0.027 | 0.059 |
| analysis_and_computation | 0.428 | 0.496 | 0.363 |
| uncertainty_and_verification | 0.010 | 0.010 | 0.008 |
| unknown | 0.512 | 0.465 | 0.566 |

*Difference $\Delta S$ (Correct − Incorrect):* [ -0.001, -0.032, +0.133, +0.001, -0.101 ]

Table 41: Stratos — Mean top–level transition matrices ($\bar{T}$). Rows are sources; columns are destinations.

| | final_answer | setup_and_retrieval | analysis_and_computation | uncertainty_and_verification | unknown |
|---|---|---|---|---|---|
| *ALL* | | | | | |
| final_answer | 0.454 | 0.099 | 0.267 | 0.180 | 0.001 |
| setup_and_retrieval | 0.130 | 0.377 | 0.400 | 0.091 | 0.002 |
| analysis_and_computation | 0.170 | 0.133 | 0.543 | 0.154 | 0.001 |
| uncertainty_and_verification | 0.110 | 0.165 | 0.349 | 0.375 | 0.000 |
| unknown | 0.231 | 0.341 | 0.352 | 0.059 | 0.017 |
| *CORRECT* | | | | | |
| final_answer | 0.456 | 0.097 | 0.257 | 0.189 | 0.001 |
| setup_and_retrieval | 0.126 | 0.353 | 0.436 | 0.084 | 0.001 |
| analysis_and_computation | 0.194 | 0.137 | 0.536 | 0.133 | 0.001 |
| uncertainty_and_verification | 0.150 | 0.178 | 0.359 | 0.312 | 0.000 |
| unknown | 0.222 | 0.361 | 0.250 | 0.083 | 0.083 |
| *INCORRECT* | | | | | |
| final_answer | 0.431 | 0.105 | 0.286 | 0.177 | 0.001 |
| setup_and_retrieval | 0.129 | 0.385 | 0.390 | 0.093 | 0.002 |
| analysis_and_computation | 0.157 | 0.133 | 0.550 | 0.160 | 0.001 |
| uncertainty_and_verification | 0.091 | 0.162 | 0.352 | 0.395 | 0.000 |
| unknown | 0.220 | 0.297 | 0.338 | 0.095 | 0.050 |

Table 42: Stratos — Difference matrix $\Delta T = \bar{T}_{\text{correct}} - \bar{T}_{\text{incorrect}}$. Positive values are more likely in correct runs.

| | final_answer | setup_and_retrieval | analysis_and_computation | uncertainty_and_verification | unknown |
|---|---|---|---|---|---|
| final_answer | **+0.025** | -0.008 | -0.029 | +0.012 | +0.000 |
| setup_and_retrieval | -0.003 | -0.032 | +0.046 | -0.010 | -0.001 |
| analysis_and_computation | +0.037 | +0.004 | -0.014 | -0.028 | -0.000 |
| uncertainty_and_verification | +0.059 | +0.016 | +0.007 | -0.083 | +0.000 |
| unknown | +0.002 | +0.065 | -0.088 | -0.012 | +0.033 |

Table 43: Stratos — Mean start distributions ($\bar{S}$) and $\Delta S = \bar{S}_{\text{correct}} - \bar{S}_{\text{incorrect}}$.

| | ALL | CORRECT | INCORRECT |
|---|---|---|---|
| final_answer | 0.000 | 0.000 | 0.000 |
| setup_and_retrieval | 0.821 | 0.760 | 0.853 |
| analysis_and_computation | 0.161 | 0.218 | 0.128 |
| uncertainty_and_verification | 0.018 | 0.022 | 0.019 |
| unknown | 0.000 | 0.000 | 0.000 |
| *Difference $\Delta S$ (Correct − Incorrect):* [ 0.000, -0.093, +0.090, +0.002, 0.000 ] | | | |

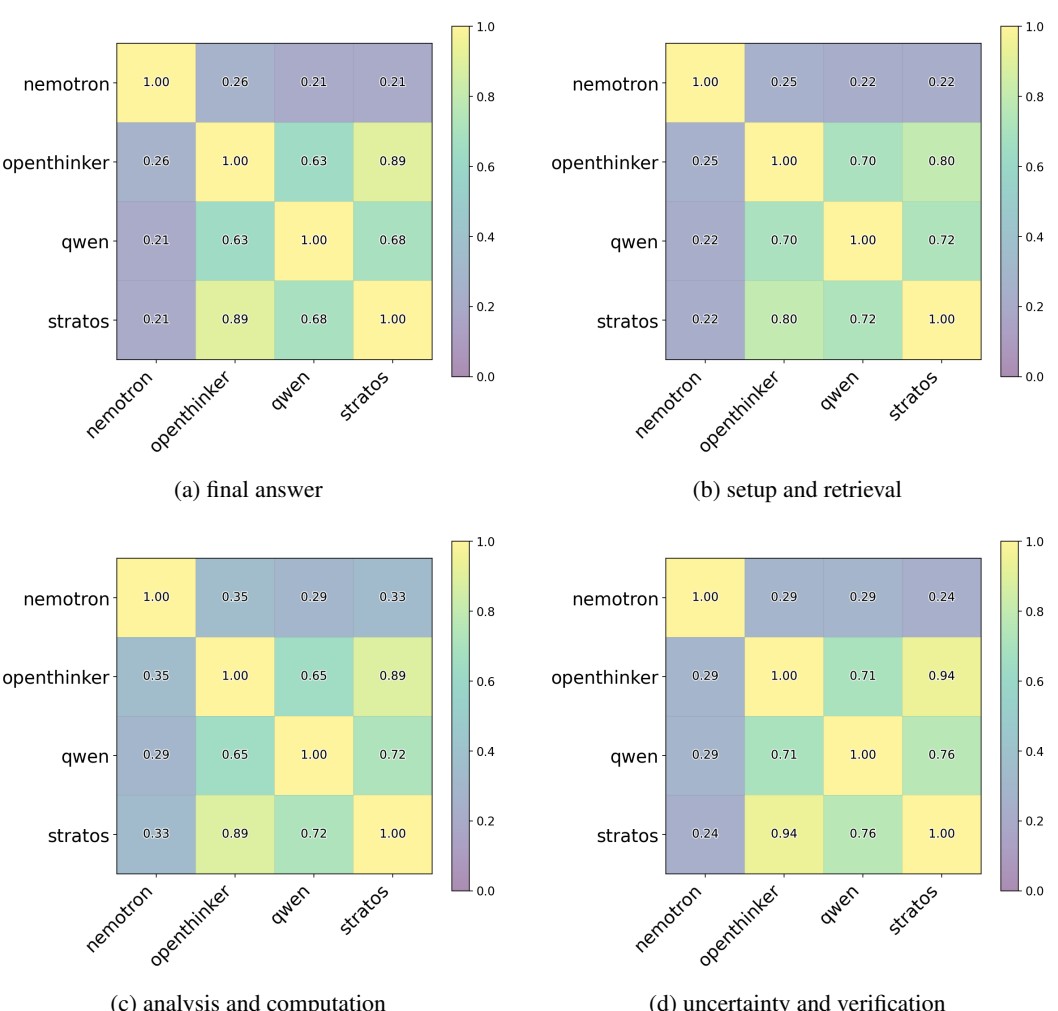

(a) final answer

(b) setup and retrieval

(c) analysis and computation

(d) uncertainty and verification

Figure 4: Percent-depth endpoint overlap.

Table 44: Per-model × category × subset consensus.

| subset | model | category_name | segments_bins |
|---|---|---|---|
| all | nemotron | analysis_and_computation | 0-32—33-54—55-57—58-81—82-96—97-99 |
| all | openthinker | analysis_and_computation | 0-27—28-30—31-65—66-75—76-82—83-85—86-92—93-96—97-99 |
| all | qwen | analysis_and_computation | 0-37—38-58—59-61—62-68—69-71—72-82—83-85—86-96—97-99 |
| all | stratos | analysis_and_computation | 0-27—28-65—66-68—69-78—79-82—83-89—90-96—97-99 |
| all | nemotron | unknown | 0-75—76-96—97-99 |
| all | openthinker | unknown | 0-65—66-78—79-96—97-99 |
| all | qwen | unknown | 0-58—59-68—69-71—72-82—83-89—90-92—93-96—97-99 |
| all | stratos | unknown | 0-65—66-78—79-82—83-99 |
| all | nemotron | final_answer | 0-29—30-54—55-81—82-84—85-96—97-99 |
| all | openthinker | final_answer | 0-27—28-68—69-82—83-85—86-99 |
| all | qwen | final_answer | 0-37—38-40—41-58—59-61—62-68—69-71—72-85—86-96—97-99 |
| all | stratos | final_answer | 0-27—28-61—62-65—66-75—76-78—79-85—86-89—90-99 |
| all | nemotron | setup_and_retrieval | 0-38—39-51—52-60—61-81—82-84—85-96—97-99 |
| all | openthinker | setup_and_retrieval | 0-68—69-71—72-82—83-85—86-99 |
| all | qwen | setup_and_retrieval | 0-37—38-58—59-61—62-68—69-78—79-85—86-96—97-99 |
| all | stratos | setup_and_retrieval | 0-27—28-65—66-75—76-78—79-85—86-89—90-99 |
| all | nemotron | uncertainty_and_verification | 0-38—39-63—64-72—73-96—97-99 |
| all | openthinker | uncertainty_and_verification | 0-33—34-65—66-68—69-75—76-78—79-85—86-89—90-99 |
| all | qwen | uncertainty_and_verification | 0-37—38-61—62-68—69-71—72-82—83-85—86-96—97-99 |
| all | stratos | uncertainty_and_verification | 0-23—24-65—66-75—76-78—79-85—86-89—90-99 |
| correct | nemotron | analysis_and_computation | 0-32—33-35—36-54—55-57—58-72—73-78—79-96—97-99 |
| correct | openthinker | analysis_and_computation | 0-27—28-65—66-75—76-82—83-85—86-89—90-96—97-99 |
| correct | qwen | analysis_and_computation | 0-37—38-58—59-61—62-68—69-78—79-82—83-85—86-96—97-99 |
| correct | stratos | analysis_and_computation | 0-20—21-30—31-68—69-78—79-82—83-85—86-89—90-96—97-99 |
| correct | nemotron | unknown | 0-38—39-75—76-99 |
| correct | qwen | unknown | 0-44—45-58—59-68—69-71—72-75—76-85—86-89—90-96—97-99 |
| correct | stratos | unknown | 0-27—28-78—79-99 |
| correct | nemotron | final_answer | 0-35—36-84—85-96—97-99 |
| correct | openthinker | final_answer | 0-23—24-61—62-68—69-78—79-82—83-85—86-99 |
| correct | qwen | final_answer | 0-37—38-58—59-61—62-68—69-71—72-85—86-96—97-99 |
| correct | stratos | final_answer | 0-27—28-65—66-68—69-78—79-82—83-89—90-99 |
| correct | nemotron | setup_and_retrieval | 0-32—33-54—55-72—73-75—76-96—97-99 |
| correct | openthinker | setup_and_retrieval | 0-27—28-30—31-65—66-68—69-82—83-85—86-99 |
| correct | qwen | setup_and_retrieval | 0-37—38-58—59-61—62-68—69-85—86-96—97-99 |
| correct | stratos | setup_and_retrieval | 0-27—28-65—66-68—69-82—83-85—86-99 |
| correct | nemotron | uncertainty_and_verification | 0-32—33-38—39-63—64-93—94-96—97-99 |
| correct | openthinker | uncertainty_and_verification | 0-27—28-65—66-68—69-75—76-82—83-85—86-99 |
| correct | qwen | uncertainty_and_verification | 0-37—38-40—41-61—62-68—69-75—76-85—86-96—97-99 |
| correct | stratos | uncertainty_and_verification | 0-23—24-27—28-65—66-71—72-78—79-85—86-99 |
| incorrect | nemotron | analysis_and_computation | 0-32—33-54—55-78—79-81—82-96—97-99 |
| incorrect | openthinker | analysis_and_computation | 0-27—28-30—31-65—66-68—69-75—76-82—83-85—86-92—93-96—97-99 |
| incorrect | qwen | analysis_and_computation | 0-37—38-58—59-61—62-68—69-71—72-85—86-96—97-99 |
| incorrect | stratos | analysis_and_computation | 0-27—28-65—66-75—76-82—83-85—86-89—90-96—97-99 |
| incorrect | nemotron | unknown | 0-35—36-38—39-96—97-99 |
| incorrect | openthinker | unknown | 0-40—41-44—45-68—69-71—72-89—90-96—97-99 |
| incorrect | qwen | unknown | 0-37—38-65—66-68—69-71—72-75—76-82—83-92—93-96—97-99 |
| incorrect | stratos | unknown | 0-78—79-99 |
| incorrect | nemotron | final_answer | 0-32—33-54—55-84—85-96—97-99 |
| incorrect | openthinker | final_answer | 0-27—28-65—66-75—76-78—79-85—86-89—90-96—97-99 |
| incorrect | qwen | final_answer | 0-37—38-40—41-58—59-61—62-68—69-71—72-75—76-82—83-85—86-96—97-99 |
| incorrect | stratos | final_answer | 0-23—24-65—66-71—72-75—76-78—79-85—86-99 |
| incorrect | nemotron | setup_and_retrieval | 0-32—33-54—55-75—76-96—97-99 |
| incorrect | openthinker | setup_and_retrieval | 0-27—28-30—31-68—69-75—76-78—79-82—83-85—86-99 |
| incorrect | qwen | setup_and_retrieval | 0-37—38-40—41-61—62-68—69-78—79-82—83-96—97-99 |
| incorrect | stratos | setup_and_retrieval | 0-27—28-65—66-68—69-75—76-82—83-85—86-89—90-99 |
| incorrect | nemotron | uncertainty_and_verification | 0-32—33-54—55-78—79-96—97-99 |
| incorrect | openthinker | uncertainty_and_verification | 0-33—34-65—66-68—69-75—76-82—83-85—86-89—90-99 |
| incorrect | qwen | uncertainty_and_verification | 0-37—38-40—41-61—62-71—72-82—83-85—86-96—97-99 |
| incorrect | stratos | uncertainty_and_verification | 0-27—28-65—66-75—76-85—86-89—90-96—97-99 |

