# OpenReview forum: "Hidden Markov Modeling of Reasoning Dynamics in Large Language Models"
_ICLR.cc/2026/Conference — Submitted to ICLR 2026_

### Official Review · Reviewer_HKvS · 2025-10-27

**Soundness:** 3
**Presentation:** 2
**Contribution:** 3
**Rating:** 4
**Confidence:** 4

**Summary:**

This paper presents a Hierarchical Hidden Markov Model (HHMM) that unifies explicit reasoning in text and implicit reasoning in hidden states. It models reasoning as semantic transitions over depth regimes and finds that successful trajectories align semantic and structural dynamics, while failures show unstable loops. Derived steering vectors enable targeted interventions that correct reasoning without increasing output length.

**Strengths:**

The HHMM provides an elegant probabilistic formalism unifying textual reasoning steps with internal representational dynamics—bridging behavior-level and mechanism-level reasoning analysis.

**Weaknesses:**

Both the correct and incorrect trajectories show strong self-loops at the analysis step (0.382 and 0.396), yet the paper provides different explanations for these similar patterns. This suggests the need for additional experiments and statistical analysis to ensure the interpretation is reliable. Similarly, when comparing the correct and incorrect transition matrices, the relative ranks of most elements appear similar. It would be better to include statistical significance tests or clearer evidence to support the claim that the transition dynamics truly differ between the two cases.

The experimental criterion — “included in the consensus set if at least two out of four models agree” — is also unconvincing. As Table 3 suggests, different models exhibit distinct reasoning strategies; thus, using shared anchors across architectures may conflate semantically different depth regions. More justification is needed to claim cross-model consensus on reasoning anchors.

Some descriptions are unclear or imprecise. For example, in Section 5, “surface-level” should likely be “top-level.” Further clarifications are needed (see questions below).

**Questions:**

What models and datasets were used to compute the statistics in Table 1? Could the authors include more datasets and LLMs to better illustrate the transition dynamics? Also, what do the colors in Table 1 represent?

In Table 2, what exactly does “retrieval” mean in the term setup_and_retrieval?

The paper poses an important question — how explicit and implicit reasoning work together in solving problems — but the conclusion remains unclear. Could the authors provide a more explicit summary or interpretation of how these two dimensions interact?

---

> ### Author Response · Authors · 2025-12-03
>
> We thank the reviewer for the careful and constructive evaluation and for recognizing the **elegance** of our framework. We also appreciate the opportunity to clarify the significance of our results, our experimental criteria, and the design of our explicit–implicit framework, as well as to improve the clarity of the paper.
>
> **Regarding results significance**
>
> Regarding the significance of the results, the key point is how these differences accumulate along the reasoning trajectory. Since both correct(successful) and incorrect(unsuccessful) runs come from the same NLG model, we expect most transitions to be similar. But these differences, though modest at individual transitions, **compound multiplicatively** across multi-step reasoning, amplifying small per-step advantages into substantial performance gaps. For example, on the analysis→analysis loop (0.493 vs. 0.512, Δ = 0.019), this gap compounds multiplicatively: over 10 consecutive analysis steps, the probabilities diverge to 0.493^{10}≈0.0067 vs. 0.512^{10}≈0.0127, making the unsuccessful trajectories nearly twice as likely to remain in analysis. This illustrates why small per-step deviations can produce substantially different long-range behaviors.
>
> To ensure these patterns are not artifacts of a particular model or dataset, Appendix G reports a bootstrap analysis. The resulting confidence intervals show that the direction of these differences is stable across runs. Even for the analysis→analysis self-loop—a relatively small numerical gap in Table 2—the bootstrap 95% CI (Δ ∈ [−0.033, −0.007]) remains strictly below zero, indicating that incorrect trajectories consistently self-loop more often than correct ones and confirming the **statistical significance** of the observed transitions.
>
> **Regarding the experimental criterion**
>
> Based on the reviewer’s suggestion, we revised Section 5 to identify anchors in a way that explicitly respects architectural differences. Our similarity analysis shows strong within-family consistency and clear between-family divergence. Motivated by this empirical structure, we group models by their families for analysis, avoiding cross-architecture pooling and the ambiguous criteria. Examining these families side-by-side also reveals higher-level patterns that persist across different model lineages.
>
> **Regarding the framework design – explicit-implicit stage of EIRL**
>
> We clarify it as a two-stage model that covers both explicit and implicit perspective and make clear what the framework is modeling and how it matches the causal chain we care about. We have revised Section 2 accordingly. Our motivation is to study how internal depth-wise computation gives rise to external reasoning behavior in text and therefore alter the outcome. EIRL is designed as a **reverse engineer** for this: the explicit stage captures how explicit reasoning roles evolve across steps, while the implicit stage captures how the model allocates computation across layers within a given role. This two-stage design therefore represents the internal to external logical chain *At the implicit stage, hidden states organize into distinct depth patterns that differ across reasoning categories, indicating that the model allocates its layers differently depending on the functional role of the step. These internal configurations then **give rise** to the explicit stage, where the model expresses its reasoning through semantic transitions.*
>
> We add an experiment (see Figure 4) that uses two stages together: we select the maximum reasoning transitions at the explicit stage and then apply steering at the EIRL-identified transition layers at the implicit stage. The results show that steering at EIRL-identified transition layers always yields better performance, indicating that both stages of the framework operate **jointly** and coherently to capture the model reasoning.
>
> **Regarding paper writing**
>
> First, we have removed the imprecise “surface-level / top-level” terminology and now consistently use “explicit” and “implicit” stages throughout the paper. Second, Table 1 average transition statistics across all models and datasets used in our study. We have clarified this in the main text and also made the color coding explicit: blue = stable self-loops, yellow = branching from setup, and green = verification loops. Finally, the “retrieval” in “setup_and_retrieval” refers to the stage where the model retrieves necessary facts from the input. We have revised our paper to improve the clarity.

---

> ### Author Response · Authors · 2025-12-03
> **Response Summary**
>
> We clarify the significance of our results and the experimental settings.
>
> (1) We show that small transition differences between correct and incorrect trajectories accumulate multiplicatively along multi-step reasoning, producing large behavioral divergences; bootstrap confidence intervals confirm that these effects are **statistically significant**.
>
> (2) Following the reviewer’s suggestion, we revise our analysis to identify anchors in a way that respects model families and examine higher-level patterns that persist across different model lineages.
>
> (3) We clarify EIRL’s design as a two-stage explicit–implicit framework that mirrors the causal chain **from depth-wise computation to external reasoning behavior**. A new experiment (Fig. 4) shows that steering at EIRL-identified transition layers yields better performance, demonstrating that both stages **jointly** capture the model’s reasoning organization.

---

### Official Review · Reviewer_J8mT · 2025-10-30

**Soundness:** 3
**Presentation:** 3
**Contribution:** 2
**Rating:** 4
**Confidence:** 4

**Summary:**

The paper models LLM reasoning as a two-level process that couples what each step is doing with how hidden states evolve across layers. It introduces a hierarchical HMM: a top chain over step semantics (setup/retrieval, analysis, verification, final answer) and a bottom HMM over latent “depth regimes” that capture layer-wise computation during each step. Using this joint model, the authors show that successful solutions follow stable semantic paths anchored by consistent late-layer patterns, while failures loop in verification or show unstable depth transitions. They then derive step-aware steering vectors from the learned transitions to gently nudge hidden states at critical moments, correcting some failures without changing weights or lengthening outputs.

**Strengths:**

Strengths

- Unified lens: Couples step semantics with hidden-state dynamics, giving a clear “reasoning trajectory.”

- Actionable diagnostics: Pinpoints where runs stall (e.g., verification loops) and which layers fail to anchor.

- Lightweight steering: Step-aware vectors nudge failing traces without changing weights or lengthening outputs.

**Weaknesses:**

Weaknesses

- Small gains: Correction improvements over baselines are modest; may feel insignificant for production.

- Label reliance: Top-level step tags are self-annotated by the model, risking bias and propagation of errors.

- Simplifying assumptions: Bottom-level uses PCA + diagonal Gaussians; may miss richer, non-Gaussian structure.

**Questions:**

same as weakness

---

> ### Author Response · Authors · 2025-12-03
>
> We thank the reviewer for the constructive assessment and for recognizing the **strength of our framework**, the **insightfulness of our analytical findings**, and our **lightweight steering mechanism**. We also appreciate the opportunity to clarify the value of the steering section, our labeling method, and the experimental settings.
>
> **Regarding the steering section**
>
> We would like to clarify that the primary contribution of EIRL is to provide an interpretability and analysis tool. The steering experiments are included **not to establish a new steering strategy** to outperform existing methods, but to **verify** that our framework successfully discovers the structural differences between successful and unsuccessful reasoning paths. We add a layer ablation study (Figure 4) and show that steering performance shows local optima at EIRL-identified transition layers. This alignment validates that EIRL successfully identifies computational control points in the model's internal reasoning process. Together, this steering section proves that the EIRL provides insights for both internal depth organization and external semantic transitions. We also provide a qualitative example in Appendix~\ref{appendix:qualitative-example} to intuitively demonstrate how steering reshapes the reasoning trajectory. After applying the steering vector that encourages uncertainty_and_verification to final_answer, the model exhibits fewer repetitive verification loops and instead transitions more decisively toward conclusions after performing verification, avoiding the excessive self-doubt patterns observed in the baseline.
>
> To further illustrate the broader utility of EIRL, we have added Section 7, which demonstrates a complementary **interpretability application** of our framework. Many existing interventions successfully improve reasoning performance or reduce overthinking, but they are typically evaluated only on aggregate metrics (accuracy, token length) without understanding how these interventions reorganize the internal reasoning flow to produce different outcomes. EIRL fills this gap by offering a diagnostic and interpretability tool. In this section, we apply EIRL to analyze different prompting intervention strategies by modeling the reasoning flow before and after intervention, revealing how each reshapes semantic transitions and explaining why they yield different accuracy and token counts.
>
> **Regarding the labelling process**
>
> We intentionally use the same model for semantic tagging because (1) we aim to capture the model’s own perception of its reasoning roles rather than impose an external interpretation, which is crucial for **faithfully** characterizing its internal dynamics and stronger external classifiers introduce distribution mismatch, (2) adding an external stronger model add significant computational cost, and (3) EIRL is robust to small tagging noise because it models aggregate transition patterns, not individual labels. Following the reviewer’s suggestion regarding classifier reliability, we also included an “unknown” category to capture cases where the classifier cannot make a confident classification, results are updated accordingly and remain consistent with our previous conclusions. This reduces labeling noise and improves the robustness and stability of the reported statistics.
>
> **Regarding experiment setting**
>
> Our use of diagonal Gaussians is not a modeling assumption about the true geometry of hidden states, but a practical parametrization of the HMM emission needed for stable regime discovery. Diagonal Gaussian emissions are the standard, numerically stable parameterization for high-dimensional HMMs [1]. EIRL is not intended to model the full nonlinear manifold of hidden representations; its goal is only to group layers into regimes and identify where structural shifts occur. For this purpose, ignoring fine-grained non-Gaussian structure is inconsequential. High dimensionality makes direct estimation unstable; PCA keeps the main structure while ensuring reliable Gaussian fitting.
>
> [1] Rabiner, Lawrence R. "A tutorial on hidden Markov models and selected applications in speech recognition." Proceedings of the IEEE 77.2 (2002): 257-286.

---

> ### Author Response · Authors · 2025-12-03
> **Response Summary**
>
> We clarify the experiment settings and provide supporting analyses to address the reviewer’s concerns.
>
> (1) We emphasize that the steering experiments are **not intended** to propose a new steering method, but to **validate** that EIRL correctly identifies the differences between successful and unsuccessful reasoning paths. Results in Section 6 proves that the EIRL provides insights for both internal depth organization and external semantic transitions.
>
> (2) We highlight EIRL’s broader **interpretability function**: we added Section 7 that demonstrates that EIRL reveals how different prompting interventions reorganize semantic transitions, explaining differences in accuracy and token length that aggregate metrics alone cannot capture.
>
> (3) We justify using the same model for semantic tagging: it preserves **faithfulness** to the model’s own reasoning semantics, avoids significant additional **cost** from external classifiers, and EIRL’s aggregate transition modeling is robust to small tagging noise.
>
> (4) We clarify that PCA and diagonal Gaussian emissions constitute standard, numerically stable HMM parameterizations, and they are fully sufficient for our objective of discovering regime shifts in model depth.

---

### Official Review · Reviewer_q8fR · 2025-10-31

**Soundness:** 2
**Presentation:** 2
**Contribution:** 2
**Rating:** 4
**Confidence:** 3

**Summary:**

The paper presents a Hierarchical Hidden Markov Model (HHMM) framework to jointly capture explicit semantic stages and implicit structural depth regimes underlying reasoning processes in large language models (LLMs). Experiments demonstrate that the approach can systematically characterize reasoning motifs. Based on HHMM, intervention can be applied to improve performance without increasing sequence length.

**Strengths:**

1. **Important Topic**: The interpretability of LLM reasoning is an important and under-explored research area.

2. **Novel and Interesting Modeling**: The introduction of HHMM to connect explicit semantic transitions with latent computational regimes is a non-trivial approach for interpreting and intervening in LLM reasoning. The analytical results offer interesting insights about reasoning behaviors of current LLMs.

3. **Actionable Steering Mechanism**: Beyond analytical experiments, this paper proposes intervention technique derived from transition matrices. The steering mechanism not only helps validate the analysis reliability, but offers a pragmatic way to rescue failing trajectories at some times.

**Weaknesses:**

1. **Limited Justification of HHMM Assumptions**: This paper lacks a formal discussion or ablation on the necessity of the hierarchical structure for capturing meaningful interaction between semantic roles and depth regimes. It is unclear the observed improvements stem from hierarchy or from simply more latent structure.

2. **Experimental Issues** Some experimental settings including models, $C$, $K$, boundary clusters, etc., seem arbitrary. Results in Table 1, 2 and the relative differences are not significant enough for the analytical claims about reasoning patterns in Section 4 Q1, Q2.

3. **Insufficient Baseline Comparison for Steering**: In Table 6, the proposed intervention technique is only compared with a basic edge-agnostic baseline, rather than established steering or latent intervention frameworks. Considering the modest intervention improvement (about 1pp), comparison with more related methods is important.

**Questions:**

1. **Necessity of Hierarchy (Weakness 1 Related)**: Could the authors experimentally or theoretically justify why the hierarchical HMM is preferable to a single-level HMM or other latent-state models? For example, would a flat HMM over step * depth suffice?

2. **Model Selection (Weakness 2 Related)** Why use a Qwen3-1.7B instead of its 4B or 8B variations, which are consistent in parameter size with other models?

3. **Robustness (Weakness 2 Related)**: How sensitive are the bottom-level HMMs to number of regimes ($K$), or the PCA preprocessing step? And other experimental variations like tasks and models? Are the results stable across runs?

---

> ### Author Response · Authors · 2025-12-03
> **Response (1/2)**
>
> We thank the reviewer for the thoughtful and constructive feedback and for recognizing the **importance of our problem setting**, **the novelty of our framework**, and the **actionability of our steering method**. We also appreciate the opportunity to clarify our experimental design and settings and to provide additional analyses addressing the reviewer’s concerns.
>
> **Regarding the framework design – explicit-implicit stage of EIRL**
>
> We clarify it as a two-stage model that covers both explicit and implicit perspective and make clear what the framework is modeling and how it matches the causal chain we care about. We have revised Section 2 accordingly. Our motivation is to study **how internal depth-wise computation gives rise to external reasoning behavior** in text and therefore alter the outcome. EIRL is designed as a reverse engineer for this: the explicit stage captures how explicit reasoning roles evolve across steps, while the implicit stage captures how the model allocates computation across layers within a given role. This two-stage design therefore represents the internal to external causal chain that our work aims to analyze.
>
> Given this setup, a “flat model over step×depth” mentioned by the reviewer is not an equivalent alternative. Although the hidden states could form as a (step size \times depth) grid, a Hidden Markov Model requires a one-dimensional sequence as input. To apply a single-level HMM over step×depth, one would have to flatten the grid into a sequence such as [h_{1,1}, h_{1,2}, ..., h_{1,L},; h_{2,1}, ..., h_{2,L},; ...; h_{T,L}], and impose a single Markov chain over this ordering. In this formulation, transitions along semantic time (between steps) and transitions along network depth (between layers) are forced to share the same latent dynamics. As a result, semantic and structural factors become entangled in uninterpretable latent states.
>
> By contrast, our two-stage EIRL explicitly factorizes these dimensions. The explicit-stage chain yields a transition matrix over reasoning types, giving a semantic-level description of how reasoning flows. The implicit-stage yields category-specific depth regimes and their transition layers, giving an internal description of how computation is organized across depth. We add an experiment (see Figure 4) that uses two stages together: we select the maximum reasoning transitions at the explicit stage and then apply steering at the EIRL-identified transition layers at the implicit stage. The results show that steering at EIRL-identified transition layers always yields better performance, indicating that both stages of the framework are essential for capturing the model’s reasoning processes. This effect cannot be explained by “simply using more latent structure”; it specifically depends on the interplay between the two stages.
>
> **Regarding Design Choice**
>
> Our categories set C are based on the patterns observed in prior work (e.g., Bogdan et al., 2025). These works provide empirical grounding that reasoning traces consist of a few distinct phases; we adapt this structure to cover the full reasoning paths. in our tasks. Using many more categories makes transitions too sparse and unstable, while using fewer categories merges distinct reasoning behaviors. For the implicit regimes K, we balance granularity and interpretability: very large K reduces to “one regime per layer”, which is uninformative, whereas very small K erases meaningful depth reallocations. Empirically, K=7 gives clear and stable boundaries, and Appendix F shows that varying K produces similar implicit pattern and steering behavior. Boundaries are identified via Viterbi decoding, retaining only those stable across runs. The revised paper groups models by architectural family, enabling family-specific and cross-family comparisons. We use Qwen-1.7B, Nemotron-4B, and Bespoke-Stratos-7B and OpenThinker-7B to ensure coverage of models at different scales.

---

> ### Author Response · Authors · 2025-12-03
> **Response (2/2)**
>
> **Regarding the significance of the results**
>
> Regarding the significance of the results, the key point is how the differences in the transition probabilities accumulate along the reasoning trajectory. Since both successful (lead to correct predictions) and unsuccessful (fail to reach correct answer) runs come from the same NLG model, we expect most transitions to be similar. But these differences, though modest at individual transitions, compound multiplicatively across multi-step reasoning, amplifying small per-step advantages into substantial performance gaps. For example, on the analysis→analysis loop (0.493 vs. 0.512, Δ = 0.019), this gap compounds multiplicatively: over 10 consecutive analysis steps, the probabilities diverge to 0.493^10≈0.0067 vs. 0.512^10≈0.0127, making the unsuccessful trajectories nearly twice as likely to remain in analysis. This illustrates why small per-step deviations can produce substantially different long-range behaviors. We have revised our paper to clarify this quantitatively.
>
> To ensure these patterns are not artifacts of a particular model or dataset, Appendix G reports a bootstrap analysis. The resulting confidence intervals show that the direction of these differences is stable across runs. Even for the analysis→analysis self-loop—a relatively small numerical gap in Table 2—the bootstrap 95% CI (Δ ∈ [−0.033, −0.007]) remains strictly below zero, indicating that unsuccessful trajectories consistently self-loop more often than successful ones and confirming the statistical significance of the observed transitions.
>
> **Regarding the baseline**
>
> We conducted an additional experiment where we directly prompted the model using two prompts: one encouraging the model to “be more concise”, and another one is a direction-guided prompt explicitly instructing the model to reason in a forward, non-looping manner.
>
> We want to emphasize that the steering experiments are included not to establish a new steering strategy to outperform existing methods, but to verify that our framework successfully discovers the structural differences between successful and unsuccessful reasoning paths.Below is the result for Bespoke-Stratos-7B on MATH-500 (seed=1):
>
> | **Method**             | **Correction Rate** | **Avg Tokens** |
> |------------------------|-----------------|------------|
> | Soft-Steering (ours)   | 0.3097          | 1820       |
> | Hard-Steering (ours)   | 0.2832          | 1838       |
> | Prompt baseline 1      | 0.2743          | 1776       |
> | Prompt baseline 2      | 0.2478          | 1859       |
>
> Both EIRL-based steering (including both Soft-Steering and Hard-Steering) methods remain competitive or superior to the prompt baselines.
>
> In addition, we have added Section 7, which demonstrates a complementary purpose of EIRL. Many existing interventions successfully attempt to improve reasoning performance or reduce overthinking. However, they are typically evaluated only on aggregate metrics (accuracy, token length), without understanding how these interventions reorganize the internal reasoning flow to therefore influence the outcome results.
>
> **EIRL fills this gap by offering a diagnostic and interpretability tool.** Instead of only checking end results, EIRL allows us to verify how each intervention reshapes the semantic transitions in the reasoning process and to explain why different strategies lead to different outcomes—by modeling the reasoning flow before and after the intervention.
>
> **Regarding ablation analysis**
>
> We vary PCA dimensions (d \in {32, 64, 128}) when constructing steering vectors, details in Appendix G. Results show that steering performance is stable across PCA dimensions. We use $d=64$ in our main experiments as it provides a balance between capturing sufficient variance and computational efficiency. Below is the PCA Ablation result for Bespoke-Stratos-7B on MATH-500 (seed=1):
>
> | **PCA Dim** | **Soft Corr.** | **Hard Corr.** | **Soft Tokens** | **Hard Tokens** |
> | ------: | ---------: | ---------: | ----------: | ----------: |
> |      32 |       0.21 |       0.25 |        1803 |        1845 |
> |      64 |       0.21 |       0.25 |        1805 |        1843 |
> |     128 |       0.21 |       0.24 |        1807 |        1841 |
>
> We also test different numbers of regimes (K = 4, 7, 10) and show that K = 7 is the best trade-off to capture meaningful transitions without over-segmentation. Across all (K), two core patterns remain stable: i) implicit stage patterns remain consistent: the delayed first boundary in verify, verify shows the lowest jaccard value. ii) EIRL-selected layers reliably correspond to local steering optima.

---

> ### Author Response · Authors · 2025-12-03
> **Response Summary**
>
> We clarify the framework design and provide new analyses to address the reviewer’s questions.
>
> (1) We clarify EIRL as a two-stage explicit–implicit framework that aligns with the causal chain **from internal depth-wise computation to external reasoning** behavior. The two-stage factorization is necessary. The flattened step×depth model suggested by the reviewer would collapse semantic and structural transitions into a single latent chain, losing interpretability.
>
> (2) We clarify why each design choice is made: model parameters balance granularity and stability, while model selection ensures coverage across scales and architectures. We provide ablation studies for key parameters in Appendix G to validate their stability.
>
> (3) We demonstrate why the observed transition differences are **significant**: even small per-step deviations compound multiplicatively along multi-step reasoning and produce large behavioral divergences; bootstrap confidence intervals confirm these effects are statistically reliable.
>
> (4) We add prompt-based baselines and EIRL-steering remains competitive, but we emphasize that steering is not meant as a new steering method—its purpose is to **validate** that EIRL correctly recovers the structural differences between successful and unsuccessful reasoning paths.
>
> (5) Finally, we include a new analysis showing that EIRL provides a complementary **interpretability** function: it reveals how interventions reorganize semantic transitions and why different strategies produce different outcomes, offering insight beyond solely accuracy or token length.

---

### Official Review · Reviewer_RnB2 · 2025-11-01

**Soundness:** 3
**Presentation:** 3
**Contribution:** 3
**Rating:** 6
**Confidence:** 3

**Summary:**

Reasoning in LLMs can be viewed as explicit and implicit steps. By explicit reasoning, the authors refer to step by step generation of text and by implicit reasoning, they refer to the changes in hidden states. By modeling explicit steps as Markov chains and implicit steps as Hidden Markov Chains they propose a new framework, HHMM (Hierarchical Hidden Markov Model), to model the reasoning trajectories in both explicit and implicit dimensions. Finally, they use their framework to steer the reasoning trajectories by adding step-aware steering vectors to the hidden states of the final layer during generation in order to guide the reasoning toward correct reasoning trajectories. Overall, they improve the reasoning for incorrect reasoning trajectories and also provide an interpretable framework to study reasoning behavior in terms of layer concentration and different aspects such as verification, analysis, and outputting the final answer.

**Strengths:**

A new framework that can discover and explain the patterns in reasoning and highlight the differences between correct and incorrect reasoning trajectories.

In addition to studying semantic reasoning and structural reasoning dynamics, they propose a steering method based on their framework that can correct the incorrect reasoning trajectories and help the generation to be stable and truthful.

While many methods for improving reasoning lead to an increase in the number of generated tokens, their approach maintains almost the same and sometimes even lower token count.

**Weaknesses:**

There are no results for the fraction of originally correct predictions becoming incorrect after steering (since in inference time we can't predict the model is going to do it wrong or correct and this experiment would help in observing the possible harmful effect of steering for already correct reasonings)

Lack of baselines in the steering section. A simple baseline like directly prompting the model to be more decisive during reasoning could suffice to show the superiority of using HHMM for steering as well.

**Questions:**

Could you use a more powerful LLM for classification (final_answer, setup_and_retrieval, …) for making sure there would be less error and mistakes in this phase?

---

> ### Author Response · Authors · 2025-12-03
>
> We thank the reviewer for the constructive feedback. We are glad that the reviewer recognizes the **novelty and value of our framework** and its **steering capability**.
>
> **Regarding the steering experiment setting**
>
> We design this experiment specifically to demonstrate that our EIRL framework correctly characterizes the differences between successful and unsuccessful reasoning transitions by constructing EIRL-derived vectors that explicitly capture the direction separating paths that lead to correct predictions (successful) from those that fail to reach correct predictions (unsuccessful).
>
> Because the purpose of these vectors is to repair unsuccessful reasoning, the meaningful metric is whether they can turn originally incorrect trajectories into correct ones—this is exactly what our reported “correction rate” measures.
>
> In contrast, measuring correct to incorrect flips is not aligned with the purpose of the intervention. Steering operates by adding a small directional perturbation to the hidden state; once we introduce any directional shift—one chosen specifically to fix failed cases—it is unavoidable that the originally correct trajectories may be nudged off their optimal direction. Importantly, this does not invalidate the method, because the vector is not designed to preserve the direction of correct to correct; it is designed to capture the “correction direction” (successful − unsuccessful). When this direction is applied universally, we do not expect it to be optimal for already-correct cases.
>
> **Regarding the baseline**
>
> We conducted an additional experiment where we directly prompted the model using two prompts: one encouraging the model to “be more concise”, and another one is a direction-guided prompt explicitly instructing the model to reason in a forward, non-looping manner. Below is the result for Bespoke-Stratos-7B on MATH-500 (seed=1):
>
> | **Method**             | **Correction Rate** | **Avg Tokens** |
> |------------------------|-----------------|------------|
> | Soft-Steering (ours)   | 0.3097          | 1820       |
> | Hard-Steering (ours)   | 0.2832          | 1838       |
> | Prompt baseline 1      | 0.2743          | 1776       |
> | Prompt baseline 2      | 0.2478          | 1859       |
>
> Both EIRL-based steering (including both Soft-Steering and Hard-Steering) methods remain competitive or superior to the prompt baselines.
>
> We also thank the reviewer for this suggestion, which inspired an additional analysis direction. In addition, Section 7 in our paper demonstrates a complementary purpose of EIRL. Many existing interventions successfully attempt to improve reasoning performance or reduce overthinking. However, they are typically evaluated only on aggregate metrics (accuracy, token length), without understanding how these interventions reorganize the internal reasoning flow to therefore influence the outcome results.
>
> ***EIRL fills this gap by offering a diagnostic and interpretability tool.*** Instead of only checking end results, EIRL allows us to verify how each intervention reshapes the semantic transitions in the reasoning process and to explain why different strategies lead to different outcomes—by modelling the reasoning flow before and after the intervention.
>
> **Regarding the classifier**
>
> Regarding the classifier, we intentionally use the same model for semantic tagging because (1) we aim to capture the model’s own perception of its reasoning roles rather than impose an external interpretation, which is crucial for faithfully characterizing its internal dynamics and stronger external classifiers introduce distribution mismatch, (2) adding an external stronger model add significant computational cost, and (3) EIRL is robust to small tagging noise because it models aggregate transition patterns, not individual labels. Following the reviewer’s suggestion regarding classifier reliability, we also included an “unknown” category to capture cases where the classifier cannot make a confident classification, results are updated accordingly and remain consistent with our previous conclusions. This reduces labeling noise and improves the robustness and stability of the reported statistics.

---

> ### Author Response · Authors · 2025-12-03
> **Response Summary**
>
> Below we clarify and resolve three issues concerning the steering metric, baseline comparisons, and classifier choice.
>
> (1) We show that correction rate is the meaningful metric because the steering vector is constructed as (correct − incorrect) specifically to use this contrastive direction to causally validate EIRL’s representation of the reasoning flow. Since the vector is optimized for correction, not for preserving already-correct states, correct→incorrect flips are **not indicative** of EIRL’s quality.
>
> (2) We add two prompt-based baselines and find that EIRL-steering remains competitive or superior. Further more, we add an analysis to demonstrate that EIRL offers a **new interpretability function**, revealing *how* interventions reorganize the reasoning flow to therefore change the output performance.
>
> (3) We justify using the same model for semantic tagging: it ensures faithfulness to the model’s own reasoning semantics, avoids extensive additional cost, and EIRL’s aggregate transition modeling is robust to small tagging noise.

---

### Author Response · Authors · 2025-12-03
**Summary for the AC**

We sincerely thank the reviewers and the AC for taking the time to review our paper and for their valuable comments and feedback.

We are encouraged that reviewers recognized: (1) the **novelty and importance** of our framework in bridging explicit and implicit reasoning analysis (RnB2, q8fR, J8mT, HKvS); (2) the important and **insightful analytical results** that reveal reasoning behaviors of current LLMs (q8fR, J8mT); and (3) the **actionable and lightweight** steering mechanism that validates our findings (RnB2, q8fR, J8mT).

We have addressed and clarified the four main concerns raised by reviewers:

**(1) Framework Design: Two-Stage Explicit–Implicit Structure.** (See details in responses to reviewers q8fR and HKvS.)

We clarify EIRL as a **two-stage explicit–implicit framework** aligned with the causal chain from internal depth-wise computation to external reasoning behavior. **Our analysis reveals a clear internal-to-external progression**: *at the implicit stage, hidden states organize into distinct depth patterns across reasoning categories; these internal configurations then give rise to the explicit stage, where the model expresses reasoning through semantic transitions*. EIRL is designed to reverse-engineer this process—tracing from external reasoning steps back to the internal structure that produces them.

We add two new experiments to validate this design. First, we provide a layer ablation study (Figure 4) to show that applying intervention at the specific layers and reasoning-type transitions identified by EIRL consistently yields better performance. This confirms that EIRL identifies meaningful steering points: intervening at these layers can guide reasoning to targeted trajectory and thereby change the final outcome. Second, we provide a qualitative example (Appendix H) that intuitively demonstrates how the intervention redirects the reasoning trajectory. Together, these experiments validate that our two-stage framework successfully captures both where (which layers) and what (which transitions) matter for reasoning, confirming the necessity and effectiveness of the explicit–implicit design.

**(2) Statistical Significance of Transition Differences.** (See details in responses to reviewers q8fR and HKvS.)

We show that our experimental results remain **statistically significant**, despite the relatively small differences in the transition probabilities. We clarify that reasoning involves many consecutive steps, so even small per-step differences compound multiplicatively—a slight difference in transition probability at each transition accumulates into a much larger divergence over the full trajectory. We also provide bootstrap analysis (Appendix G) confirming that these observed differences are statistically reliable across models, datasets, and random seeds.

**(3) Baseline Comparisons for Steering**. (See details in responses to RnB2 and q8fR.)

We clarify that the steering experiment in this paper is **not meant to propose a new steering method**. Its purpose is to causally validate that EIRL correctly identifies the structural differences between successful and unsuccessful reasoning paths. Nevertheless, in response to reviewers’ suggestions, we have added two prompt-based baselines to provide additional comparisons; initial results (Section 7) show that EIRL-steering performs competitively.

**(4) Broader Interpretability Tool.** (See details in responses to RnB2, q8fR, and J8mT.)

Beyond steering, we highlight EIRL's value as a diagnostic and interpretability tool. Many existing interventions are evaluated only on aggregate metrics like accuracy or token count—but these metrics alone do not explain why different strategies produce different outcomes. EIRL fills this gap. In Section 7, we show that EIRL can be applied to analyze different intervention strategies and reveal **how each one reorganizes the semantic flow of reasoning to produce its observed effects**. This demonstrates that EIRL can serve as an interpretability tool for understanding the underlying reasoning process.

---

> ### Author Response · Authors · 2025-12-03
>
> We also addressed the following minor concerns:
>
> (1) Refined cross-model consensus analysis, enabling family-specific and cross-family comparisons. (HKvS)
>
> (2) Clarified metric design and experimental settings. (RnB2, q8fR)
>
> (3) Justified same-model tagging for faithfulness to the model's own reasoning semantics; following the reviewer’s suggestion regarding classifier reliability, we also included an “unknown” category to capture cases where the classifier cannot make a confident classification, results are updated accordingly and remain consistent with our previous conclusions. This reduces labeling noise and improves the robustness and stability of the reported statistics. (RnB2, J8mT)
>
> (4) Justified modeling configurations; provided ablations on key parameters showing stable results. (q8fR, J8mT)
>
> Finally, to improve aesthetics and clarity, we rename the framework from Hierarchical Hidden Markov Model (HHMM) to Explicit–Implicit Reasoning Lens (EIRL), which better reflects its explicit–implicit structure and avoids potentially confusing terminology such as "top-down". We have updated it in the paper and will use EIRL throughout the following responses.

---

### Meta-Review · Area_Chair_1yju · 2026-01-07

**Summary:**

The submission introduces a framework employing a Hierarchical Hidden Markov Model to link explicit semantic reasoning roles with implicit latent depth regimes. While technically sophisticated, the work faces significant methodological and practical challenges. Specifically, the reliance on self-tagging for the top-level chain raises concerns regarding circular bias, and the numerical differences in transition probabilities are seen as marginal. Furthermore, the high engineering complexity yields only modest improvements over simpler baselines. Consequently, the submission does not meet the requirements for acceptance.

**Reviewer Concerns:**

Addressed:
- The authors incorporated prompt-based baselines to compare against their steering mechanism and provided bootstrap analysis to verify the statistical stability of observed transition probabilities.
- The framework was renamed to better reflect its design, and the cross-model comparison criteria were adjusted to respect specific architectural families.

Outstanding:
- Concerns persist regarding the reliance on the model to self-annotate its reasoning roles, which may lead to a biased interpretation of internal dynamics based on the model's own generation errors.
- The absolute differences in transition probabilities are numerically marginal, making it difficult to definitively separate fundamental reasoning shifts from generative noise.
- The substantial overhead required to implement the HHMM-based pipeline is not yet sufficiently justified by the modest improvements observed over simpler prompting or curation methods.

**Reviewer Scores:**

RnB2: The additional baselines aligned with the reviewer's initial assessment, resulting in a marginal accept as the final score.

q8fR: Skepticism regarding the framework's high complexity relative to its modest empirical gains remained the standing evaluation.

J8mT: Foundational concerns regarding the reliability of self-annotated labels and the practical significance of the findings persisted as the final judgment.

HKvS: The initial critique regarding the marginal numerical significance of transition patterns remained the decisive factor in the final consensus.

---

### Decision · Program_Chairs · 2026-01-26

Reject